# Mechanisms of QT prolongation by buprenorphine cannot be explained by direct hERG channel block

Phu N. Tran[1,2], Jiansong Sheng[1,3], Aaron L. Randolph[1], Claudia Alvarez Baron[1], Nicolas Thiebaud[1,4], Ming Ren[1], Min Wu[1,2], Lars Johannesen[5], Donna A. Volpe[1], Dakshesh Patel[6,7], Ksenia Blinova[6], David G. Strauss[1], Wendy W. Wu[1]*

1 Division of Applied Regulatory Science, Office of Clinical Pharmacology, Center for Drug Evaluation and Research, US Food and Drug Administration, Silver Spring, Maryland, United States of America, 2 Division of Immunology and Hematology Devices, Center for Devices and Radiological Health, US Food and Drug Administration. Silver Spring, Maryland, United States of America, 3 CiPALab, Gaithersburg, Maryland, United States of America, 4 Vertex Pharmaceuticals (Europe) Ltd, Abingdon, Oxfordshire, United Kingdom, 5 Division of Cardiology and Nephrology, Office of Cardiology, Hematology, Endocrinology and Nephrology, Office of New Drugs, Center for Drug Evaluation and Research, US Food and Drug Administration, Silver Spring, Maryland, United States of America, 6 Division of Biomedical Physics, Office of Science and Engineering Laboratories, Center for Devices and Radiological Health, US Food and Drug Administration, Silver Spring, Maryland, United States of America, 7 Cardiovascular Research Center, Massachusetts General Hospital, Harvard Medical School, Boston, Massachusetts, United States of America

* wendy.wu@fda.hhs.gov

**Data Availability Statement:** Data are available from the Open Science Framework: https://osf.io/tjuev.

## Abstract

Buprenorphine is a μ-opioid receptor (MOR) partial agonist used to manage pain and addiction. $QT_C$ prolongation that crosses the 10 msec threshold of regulatory concern was observed at a supratherapeutic dose in two thorough QT studies for the transdermal buprenorphine product BUTRANS®. Because $QT_C$ prolongation can be associated with Torsades de Pointes (TdP), a rare but potentially fatal ventricular arrhythmia, these results have led to further investigation of the electrophysiological effects of buprenorphine. Drug-induced $QT_C$ prolongation and TdP are most commonly caused by acute inhibition of hERG current ($I_{hERG}$) that contribute to the repolarizing phase of the ventricular action potentials (APs). Concomitant inhibition of inward late $Na^+$ ($I_{NaL}$) and/or L-type $Ca^{2+}$ ($I_{CaL}$) current can offer some protection against proarrhythmia. Therefore, we characterized the effects of buprenorphine and its major metabolite norbuprenorphine on cardiac hERG, $Ca^{2+}$, and $Na^+$ ion channels, as well as cardiac APs. For comparison, methadone, a MOR agonist associated with $QT_C$ prolongation and high TdP risk, and naltrexone and naloxone, two opioid receptor antagonists, were also studied. Whole cell recordings were performed at 37°C on cells stably expressing hERG, $Ca_V1.2$, and $Na_V1.5$ proteins. Microelectrode array (MEA) recordings were made on human induced pluripotent stem cell-derived cardiomyocytes (iPSC-CMs). The results showed that buprenorphine, norbuprenorphine, naltrexone, and naloxone had no effect on $I_{hERG}$, $I_{CaL}$, $I_{NaL}$, and peak $Na^+$ current ($I_{NaP}$) at clinically relevant concentrations. In contrast, methadone inhibited $I_{hERG}$, $I_{CaL}$, and $I_{NaL}$. Experiments on iPSC-CMs showed a lack of effect for buprenorphine, norbuprenorphine, naltrexone, and naloxone, and delayed repolarization for methadone at clinically relevant concentrations. The

**Funding:** This work was supported by internal funding from the United States Food and Drug Administration (FDA) and appointment to the Research Participation Programs at the Oak Ridge Institute for Science and Education (ORISE) through an interagency agreement between the Department of Energy and FDA.

**Competing interests:** JS and NT are currently employed by commercial companies. Both contributed to data collection and analysis for studies in this manuscript while participating in the ORISE Research Participation Programs. ORISE fellowship stipend was funded by the FDA. No commercial company played a role in the study design, data collection and analysis, decision to publish, and preparation of the manuscript, nor did any commercial company provide financial support of any kind for this study. Commercial affiliation through former ORISE fellowship participants' current appointment in commercial companies does not alter our adherence to all PLOS ONE policies on sharing data and materials.

mechanism of $QT_C$ prolongation is opioid moiety-specific. This remains undefined for buprenorphine, while for methadone it involves direct hERG channel block. There is no evidence that buprenorphine use is associated with TdP. Whether this lack of TdP risk can be generalized to other drugs with $QT_C$ prolongation not mediated by acute hERG channel block warrants further study.

## Introduction

Buprenorphine is a μ-opioid receptor (MOR) partial agonist used to manage opioid use disorder and pain severe enough to warrant use of opioid analgesics. In healthy volunteers, multi-day exposure to a transdermal formulation BUTRANS® has been linked with dose-dependent $QT_C$ prolongation that reaches the threshold of regulatory concern (10 ms) for the supratherapeutic dose [1]. This effect was not observed when buprenorphine was co-administered with naltrexone, an opioid receptor antagonist. Because there are buprenorphine products that achieve higher systemic exposure levels at recommended doses than the supratherapeutic dose in the BUTRANS® study, and because $QT_C$ prolongation can be associated with a rare but potentially fatal ventricular tachyarrhythmia called Torsades de Pointes (TdP), these results have led to questions about whether buprenorphine's QTc prolongation is associated with any risk of TdP.

The most common cause of drug-induced $QT_C$ prolongation and TdP is acute inhibition of hERG current ($I_{hERG}$) that is important for the repolarizing phase of the ventricular action potential (AP) [2, 3]. Examples of drugs that prolong the $QT_C$ interval yet are associated with low TdP risk (i.e., ranolazine, amiodarone, and verapamil) suggest that concomitant inhibition of late $Na^+$ ($I_{NaL}$) [4] and/or L-type $Ca^{2+}$ current ($I_{CaL}$) [5, 6] could offset the TdP risk imposed by hERG channel block. Though less common, some drugs prolong the $QT_C$ interval by increasing $I_{NaL}$ [7, 8]. Basic science studies also show that delayed repolarization and ventricular arrhythmias can occur by pharmacologically enhancing $I_{CaL}$ [9]. Knowledge of drug effects on inward and outward ventricular ionic currents is thus important for proarrhythmia risk assessment. Buprenorphine has been reported to inhibit $I_{hERG}$ [10] and peak $Na^+$ current ($I_{NaP}$) [11] in the micromolar to tens of micromolar range. These concentrations far exceed the subnanomolar free drug level achieved by therapeutic dosing. However, the studies were performed at room temperature. Given that potencies of some hERG channel blockers are temperature-sensitive [12–17], it is possible that buprenorphine's potencies against $I_{hERG}$ and $I_{NaP}$ would be different if experiments were conducted at a physiological temperature. The hERG data mentioned above were obtained using hERG1a-overexpressing cells [10]. In human ventricular myocytes, hERG channels are formed by heteromeric assembly of hERG1a and hERG1b subunits [18]. Some drugs exhibit different potencies in blocking homomeric (hERG1a) and heteromeric hERG channels [19], and buprenorphine's effect on heteromeric hERG channels has not been assessed. Additionally, a literature search produced no information regarding buprenorphine's effects on $I_{CaL}$ and $I_{NaL}$, nor multi-ion channel pharmacology data for norbuprenorphine, the major active metabolite of buprenorphine, that is also a MOR partial agonist.

To address the abovementioned knowledge gap, we performed patch clamp studies at 37˚C to characterize the effects of buprenorphine and norbuprenorphine on $I_{hERG}$ mediated by hERG1a and hERG1a/1b channels, $I_{CaL}$, $I_{NaL}$, and $I_{NaP}$ using overexpression cell lines, and microelectrode array (MEA) experiments to study drug effects on the electrical behavior of human induced pluripotent stem cell-derived cardiomyocytes (iPSC-CMs). For comparison, methadone, a MOR receptor full agonist that is associated with $QT_C$ prolongation and high TdP risk, and two opioid receptor antagonists, naltrexone and naloxone, were also studied.

For data transparency, the original patch clamp records or individual data points required to construct graphs presented in the figures are available for download at: https://osf.io/tjuev/.

## Materials and methods

### Cell lines

For $I_{hERG}$ recordings, two HEK293 cell lines were used: one stably expressing the hERG1a sub-unit to form homomeric hERG channels [20]; and the other stably expressing hERG1a and hERG1b subunits to form heteromeric hERG channels [21]. Both cell lines were provided by Dr. Gail Robertson at the University of Wisconsin-Madison. The hERG1a and hERG1a/1b cells were at approximately 75% confluence when passaged using trypsin (0.25% Trypsin-EDTA, 25200–056, Gibco), and seeded at a density of $5X10^4$—$1X10^5$ cells/mL onto 12-mm sterilized and uncoated glass coverslips (12-545-80, Fisher Scientific) housed in 35X10 mm petri dishes (430588, Corning) containing 2 mL of Dulbecco's Modified Eagle's Medium (DMEM, 30–2002, ATCC) supplemented with 10% fetal bovine serum (FBS, 10437–028, Gibco) and G-418 (100 μg/mL; 11811–031, Gibco). For hERG1a/1b cells, puromycin (0.25 μg/mL; P9620, Sigma) was also added. Following passage, hERG1a cells were incubated at 5% $CO_2$ and 37˚C for 24–48 hours prior to being used for electrophysiology studies. For the hERG1a/1b cells, doxycycline (100 ng/mL; D3072, Sigma) was added 24–48 hours prior to electrophysiology experiments to induce expression of the hERG1b subunit.

For $I_{CaL}$ recordings, a CHO cell line stably expressing human $Ca_V1.2$ α and auxiliary modulatory subunits $β_2$ and $α_2δ1$ were used (Charles River Laboratory). The cells were cultured in Ham's F12 with L-glutamine nutrient mixture (11765–054, Gibco), supplemented with tetracycline-free 10% FBS (Hyclone), 0.01 mg/mL Blasticidin S (A11139-03, Gibco), 0.25 mg/mL G-418 (10131–035, Gibco), 0.25 mg/mL Hygromycin B (10687010, Invitrogen, or 30-240-CR, Corning), and 0.40 mg/mL Zeocin (46–0509, Invitrogen) at 5% $CO_2$ and 37˚C following passage. For patch clamp recording, cells in a ~70% confluent flask were washed with D-PBS (14190–144, Gibco) and then detached using Accutase (A11105-01, Gibco, or A6964, Sigma). The detached cells were seeded onto 12 mm sterile and uncoated glass coverslips in 35X10 mm petri dishes containing 2 mL of Ham's F12 media supplemented with only 10% FBS at a density of $1X10^5$—$2X10^5$ cells/mL. At least 16 hours prior to the electrophysiology experiment, $Ca_V1.2$ expression was induced by adding 0.5 μg/mL of tetracycline hydrochloride (T7660, Sigma) to the petri dish containing the seeded cells.

For $I_{NaP}$ and $I_{NaL}$ recordings, a HEK293 cell line stably expressing human $Na_V1.5$ α and β1 subunits was used (Catalog #SB-HEK-hNa$_V$1.5; SB Drug Discovery). The cells were maintained at 5% $CO_2$ and 37˚C in complete DMEM (which is DMEM (30–2002, ATCC) supplemented with 10% FBS (30–2020, ATCC)), 600 μg/mL G-418 sulfate (30-234-CI, Corning), and 4 μL/mL Blasticidin S (A11139-03, Gibco). Cells were passaged at <90% confluence by first washing with D-PBS (14190–144, Gibco), then detaching with 1–1.5 mL of TrypLE Express (12604–13, Gibco) at 37˚C for 2–3 min. The detached cells were resuspended in DMEM and seeded onto 12 mm uncoated sterile glass coverslips housed in 35X10 mm petri dishes at a density of approximately $5X10^4$—$1X10^5$ cells/mL per dish that contained 2 mL of complete DMEM. Cells were incubated at 5% $CO_2$ and 37˚C for 24–48 hours prior to use for electrophysiology studies.

The detailed cell culture procedures for overexpression cell lines can be found at: https://osf.io/tjuev/.

### Whole cell voltage clamp electrophysiology

All experiments were conducted at 37˚C. Glass coverslips with attached cells were placed in a recording chamber mounted on an inverted (Zeiss Axiovert 135TV) or an upright microscope

(Zeiss AxioExaminer D1). The recording chamber was continuously perfused with external solution flowing at a rate of 2–3 mL/min. Recordings were made using borosilicate glass pipettes (BF150-86-10; Sutter Instrument, CA) pulled with a micropipette puller (P97; Sutter Instrument, CA) to 1.5–3 MΩ resistance when filled with internal solution.

For $I_{hERG}$, the extracellular solution contained (in mM): 130 NaCl, 10 HEPES, 5 KCl, 1 $MgCl_2 \cdot 6H_2O$, 1 $CaCl_2 \cdot 2H_2O$, and 12.5 dextrose; pH was adjusted to 7.4 with 5 M NaOH. The internal solution contained (in mM): 120 K-gluconate, 20 KCl, 10 HEPES, 5 EGTA, and 1.5 MgATP; pH was adjusted to 7.3 with 1 M KOH; ~280 mOsM. The voltage command values were corrected for the 15 mV liquid junction potential (calculated using pClamp10 software; Molecular Device, CA) that resulted from using these solutions. Cells were depolarized from a holding potential of -80 mV to +40 mV for 2 s, then repolarized to -50 mV for 1.5 s to evoke the tail current that is predominantly mediated by hERG channels, and finally returned to -80 mV. A 100 ms hyperpolarizing step from -80 mV to -90 mV was included prior to the depolarizing step to monitor input resistance throughout the recording. This voltage protocol was repeated at 15 s intervals.

For $I_{CaL}$, the external solution contained (in mM): 137 NaCl, 4 KCl, 1.8 $CaCl_2$, 1 $MgCl_2$, 10 HEPES, and 10 dextrose; pH was adjusted to 7.4 with 5M NaOH. The intracellular solution contained (in mM): 120 aspartic acid, 120 CsOH, 10 CsCl, 10 EGTA, 5 MgATP, 0.4 TrisGTP, and 10 HEPES; pH was adjusted to 7.2 with 5M CsOH; ~290 mOsM. The voltage command values were corrected for the 17 mV liquid junction potential. Cells were held at -80 mV, depolarized to 0 mV for 40 ms, further depolarized to +30 mV for 200 ms, and then ramped down to -80 mV in 100 ms (-1.1 V/s). A 100 ms hyperpolarizing step from -80 to -90 mV was included prior to the first depolarizing step to monitor input resistance throughout the recording. This voltage protocol was presented at 5 s intervals.

For $I_{NaP}$ and $I_{NaL}$ recordings, the external solution contained (in mM): 130 NaCl, 4 CsCl, 2 $CaCl_2$, 1 $MgCl_2$, 10 HEPES, and 10 dextrose; pH was adjusted to 7.4 with NaOH. The internal solution contained (in mM): 130 CsCl, 7 NaCl, 1 $MgCl_2$, 5 EGTA, and 5 HEPES; pH was adjusted to 7.2 with CsOH; ~280 mOsM. When recording $I_{NaL}$, 150 nM ATX-II was added to the external solution to slow $Na^+$ channel inactivation, thereby inducing a pronounced late component. Cells were first hyperpolarized from -95 mV to -120 mV for 200 ms to facilitate recovery of $Na^+$ channels from inactivation, then depolarized to -15 mV for 40 ms, then +40 mV for 200 ms, then ramped down to -95 mV in 100 ms (-1.35 V/s). This voltage protocol was repeated at 12.5 s intervals.

These voltage protocols were continuously presented at the respective intervals throughout the duration of the recordings. Recordings were obtained first in control solution until the amplitude of the ionic current being studied reached stability, then drug solution was bath-applied and recording repeated in the presence of drug until a new steady state current amplitude was reached. For $I_{CaL}$, current rundown upon whole cell formation exhibited multiple phases. Stability for $I_{CaL}$ experiments was thus defined as rundown reaching a steady and slower rate. Each cell was exposed to one or more concentrations of the same drug as long as membrane properties (i.e., input resistance and holding current at rest) and recording quality (current signal) remained stable.

Cells were visualized using phase contrast method for the inverted microscope and differential interference contrast-infrared (DIC-IR) method for the upright microscope. Temperatures of the in-line solution heater and recording chamber were maintained with a dual channel temperature controller (TC2BIP from Cell MicroControls for the inverted microscope setup; TC-344C from Warner Instruments for the upright microscope setup), and temperature of the perfusate near the cells was recorded throughout the experiment with a thermistor. Recordings were obtained using Multiclamp 700B amplifiers (Molecular Devices, CA). For

$I_{hERG}$, signals were filtered at 2.2 kHz and digitized at 5 kHz using a Digidata 1550A interface (Molecular Devices, CA), and transferred to a computer using pClamp10 software (Molecular Devices, CA). For $I_{CaL}$, signals were filtered at 3 kHz and digitized at 10 kHz. For $I_{NaP}$ and $I_{NaL}$, signals were filtered at 10 kHz and digitized at 20 kHz. Seal resistance was always >1 GΩ, and series resistance was electronically compensated at 80%. To allow for adequate internal solution dialysis prior to actual current recording, following whole cell formation cells were given ~2 min resting period, during which the recording and membrane properties were monitored using the "membrane test" function of the pClamp10 software.

## Data and statistical analysis for whole cell voltage clamp experiments

Current traces were analyzed using custom macros written in Igor Pro 6 (WaveMetrics) and pClamp10 software (Molecular Devices, CA). For $I_{hERG}$ and $I_{NaP}$, the peak current amplitude was subtracted against baseline current amplitude measured at the beginning of the voltage protocol (-80 mV for $I_{hERG}$ and -95 mV for $I_{NaP}$ experiments). For $I_{CaL}$ and $I_{NaL}$, the current amplitude for each recorded trace was subtracted against calculated passive current for that trace as previously described [22]. Passive current calculation was done by first using Ohm's Law to calculate input resistance using the current evoked by the -10 mV hyperpolarizing step (from -80 mV to -90 mV) for the $I_{CaL}$ voltage protocol or the -25 mV hyperpolarizing step (from -95 mV to -120 mV) for the $I_{NaL}$ voltage protocol, and then using the input resistance value and Ohm's law to calculate the passive current expected at various voltages. In a subset of $Ca_V1.2$ cells, despite the use of $Cs^+$-based internal solution, an endogenous outward current was revealed following $I_{CaL}$ inhibition. For these cells, traces obtained in the presence of 100 μM verapamil (with endogenous current present) were used for subtraction for all recorded traces to obtain $I_{CaL}$ amplitude. In some cells tested with 100 to 300 μM methadone, verapamil was not applied and complete abolition of $I_{CaL}$ was achieved. When endogenous outward current was found for these cells, traces obtained in methadone were used for subtraction to calculate $I_{CaL}$ amplitude.

The potencies of tested drugs against various ionic currents were quantified by constructing concentration-inhibition plots. For $I_{hERG}$ recordings, fractional inhibition from individual cells was calculated by first normalizing the average current amplitude of the last 3 consecutively recorded traces in drug solution to that obtained in control solution, and then subtracting this value from unity. For $I_{CaL}$ recordings, the averaged current amplitude of 10 consecutively recorded traces in drug and control solutions were used. For $I_{NaL}$ and $I_{NaP}$ recordings, the averaged values of 5 consecutively recorded traces in drug and control solutions were used. For each current, the fractional inhibition values from all cells were then pooled and plotted against drug concentrations to generate concentration-inhibition plots, and these plots were fit with the Hill equation using Igor Pro 6 (WaveMetrics, Portland, OR, USA), with minimum and maximum fractional block constrained to 0 and 1, respectively. Half inhibitory concentration ($IC_{50}$) and the Hill coefficient ($n_H$) obtained from the fit are presented as mean ± standard deviation (SD) in the text. Summary data points in the figures are shown as mean ± standard error of the mean (SEM). Statistical analysis was performed on the concentration-inhibition plots for buprenorphine on hERG1a and hERG1a/1b channels using GraphPad Prism version 8.3. Individual datapoints for each concentration were fitted to the Hill equation with variable slope, in the form of "Fractional block = $1 / (1+ (IC_{50}/[buprenorphine])^{nH})$", and the best-fit values for the $IC_{50}$ and $n_H$ were compared using the extra-sum-of-squares F test with the significance P value set to < 0.05. These data and statistical analysis plans comply with the recommendations on experimental design and analysis in pharmacology [23].

## Cell culture for human iPSC-CMs and MEA plating

iCell cardiomyocytes[2] were purchased from Fujifilm Cellular Dynamics (cat. # R1017; FCDI, Madison, Wisconsin). These cells were thawed, centrifuged at 180 g, and seeded at 50,000 cells/well in 5 μL droplets on the MEA 48-well plates (cat. # M768-KAP-48, Axion Biosystems). Just prior to use, the MEA plates were coated with fibronectin (1:20 with DPBS; cat. # 11080938001,1mg/mL, Roche) for 1 hr at 37˚C. After 1 hr, 300 μL of maintenance media was slowly added to each well in the plate. Cells on MEA plates were maintained at 37˚C and 5% $CO_2$ according to published procedures [24].

## MEA recordings

All drug testing was performed on day 7 of plating cells on the MEA plates. One day prior to the experiment, iPSC-CMs were fed with fresh maintenance media. On the day of the experiment, 1000X stock solution in DMSO was prepared for each tested drug concentration. These 1000X stocks were then made into 10X stocks by diluting with maintenance media. Final concentration of DMSO did not exceed 0.1%. DMSO (0.1%) and vehicle controls were randomly distributed across the plate to normalize for edge-effects on the multi-well MEA Maestro system (Axion Biosystems). Separate drug-dosing plates were prepared and equilibrated at 37˚C and 5% $CO_2$. MEA plates equilibrated on the Maestro for 20 min before baseline recordings. The drugs were then added by pipetting 30 μL of the 10X stock solution into each well containing 270 μL of the medium. Post-drug data were collected 30 min after drug application. Field potentials in iPSC-CMs were recorded using AxIS software version 2.4.2 (Axion Biosystems) in cardiac standard configuration (130X gain) with a sampling frequency of 12.5 kHz, and a band pass filter of 0.1–2000 Hz. A statistics compiler tool filtered the beats with spike amplitude (SA) > 0.3 mV, beat-to-beat field potential duration (FPD) consistency within 2X SD, electrode FPD consistency (10% coefficient of variation), and well FPD consistency within 2X median absolute deviation. Thirty stable beats from the last 5 min of each recording were selected for analysis using Axis CiPA analysis tool (version 1.2.3).

## Data analysis for MEA recordings

The following data inclusion criteria were applied to the baseline parameters: 1) iPSC-CM baseline spontaneous beating rate within 20–90 beats per minute (0.3–1.5 Hz); 2) baseline beating rate within 6 SDs calculated from the baseline beating rate for all wells on a given plate; 3) coefficient of variation for the baseline beat period (BP) less than 5%; and 4) SA > 0.3 mV. SA is defined as the peak-to-peak amplitude of the depolarization spike. These inclusion criteria were adopted from a published study [25].

FPD of the spontaneously-beating iPSC-CMs were corrected using Fridericia's correction formula, $FPD_C = FPD/\sqrt[3]{BP}$, where BP is iPSC-CM beat period (averaged time between consecutive depolarization spikes in MEA recordings). Double delta FPD (ΔΔFPD), double-delta beat period (ΔΔBP), and double delta spike amplitude (ΔΔSA) were calculated after correcting drug-induced FPD with plate-specific vehicle (as DMSO control) and well-based baseline FPD values. ΔΔFPD data were excluded from analysis if more than 50% of wells for a given drug concentration exhibited arrhythmia-like events.

## Drugs

The following drugs and solvent were purchased from Sigma-Aldrich: buprenorphine hydrochloride (B9275, USDEA C-III), (±)-methadone hydrochloride (M0267, USDEA C-II), naltrexone hydrochloride (N3136), and DMSO (D8418). Naloxone hydrochloride (0599),

(±)-verapamil hydrochloride (0654), E-4031 dihydrochloride (1808), and tetrodotoxin citrate or TTX (1069) were purchased from Tocris Bioscience. Norbuprenorphine hydrochloride was purchased from Noramco. ATX-II (STA-700) was purchased from Alomone Labs. To make stock solutions for patch clamp experiments, TTX, verapamil, E-4031, ATX-II, naloxone, and naltrexone were dissolved in water. Buprenorphine, norbuprenorphine, and methadone were dissolved in DMSO. When DMSO was used as a solvent to prepare stock solution, the final DMSO concentration applied to overexpression cells was $\leq$0.3% for $Ca_V1.2$ cells and $\leq$0.1% for the rest. In iPSC-CM experiments, stock solutions were all made with DMSO, and the final DMSO concentration applied to the cells was $\leq$0.1%. Aliquoted stock solutions were stored at -20˚C until the day of experimentation and were thawed, vortexed, and diluted to specific test concentrations in extracellular solution.

**Clinically relevant concentrations of the tested drugs.** The $IC_{50}$s obtained for individual drugs were compared with the maximum systemic concentrations of free or total drug following therapeutic dosing (free or total $C_{max}$, respectively) to assess the likelihood of acute cardiac ion channel block when these drugs are used as indicated. The preference for this comparison is free $C_{max}$ if the percent of plasma protein binding is known, since free drug molecules are thought to be the ones that could interact with cardiac ion channels. $C_{max}$ values used in this paper reflect the highest across all drug products for a specific moiety to reflect the highest therapeutic exposure level. For buprenorphine, free $C_{max}$ was calculated using data from the drug label for Sublocade™. Following the $4^{th}$ monthly dose of subcutaneous injection, the steady state plasma buprenorphine level reached 10.12 ng/mL, or 0.02 µM converted using a molecular weight of 467.64 g/mol. Given 96% plasma protein binding, free $C_{max}$ for buprenorphine was 0.00087 µM. For norbuprenorphine, total $C_{max}$ was based on the data after the $7^{th}$ sublingual, 24 mg dose of Subutex™ (https://www.ema.europa.eu/en/documents/assessment-report/buvidal-epar-public-assessment-report_en.pdf). Total $C_{max}$ was 9.29 ng/mL or 0.022 µM, converted using a molecular weight of 413.55 g/mol. Free $C_{max}$ was not calculated as the percent of plasma protein binding for norbuprenorphine has not been empirically determined. In-house simulation using GastroPlus package (Simulation Plus Inc.) predicted this metabolite to be extensively plasma protein-bound (95 ± 5%), just like the parent molecule. For methadone, free $C_{max}$ was calculated using a total $C_{max}$ of 904 ng/mL from Table 4 of the study by Florian and colleagues [26]. Given 85% plasma protein binding, free $C_{max}$ of methadone was 135.6 ng/mL or 0.44 µM, converted using a molecular weight of 309.45 g/mol. For naltrexone, total $C_{max}$ was based on 380 mg intramuscular injection of Vivitrol® every 4 weeks as the recommended dose (https://www.accessdata.fda.gov/drugsatfda_docs/label/2019/021897s045lbl.pdf). Total $C_{max}$ was 12.1 ng/mL or 0.035 µM when converted using a molecular weight of 341.41 g/mol. Given 21% plasma protein binding, free $C_{max}$ was 0.028 µM. For naloxone, total $C_{max}$ was the result of a 2 mg EVZIO® intramuscular/subcutaneous injection (https://www.accessdata.fda.gov/drugsatfda_docs/label/2016/209862lbl.pdf). Total $C_{max}$ was 7.91 ng/mL or 0.024 µM, converted using a molecular weight of 327.27 g/mol. In adult human plasma, 54% naloxone is in free, unbound form [27]. Thus, free $C_{max}$ of naloxone was 0.013 µM.

# Results

## Cardiac ion channel electrophysiology

Fig 1A shows a schematic diagram of a ventricular AP and contributions of the ionic currents examined in this study to different phases of the AP. In a highly simplified scheme, $I_{NaP}$ is responsible for AP upstroke (phase 0), $I_{CaL}$ and $I_{NaL}$ contribute to AP duration (APD; phase 2), and $I_{hERG}$, thought to be the correlate of $I_{Kr}$ in cardiomyocytes, contributes to repolarization of the membrane potential (phase 3).

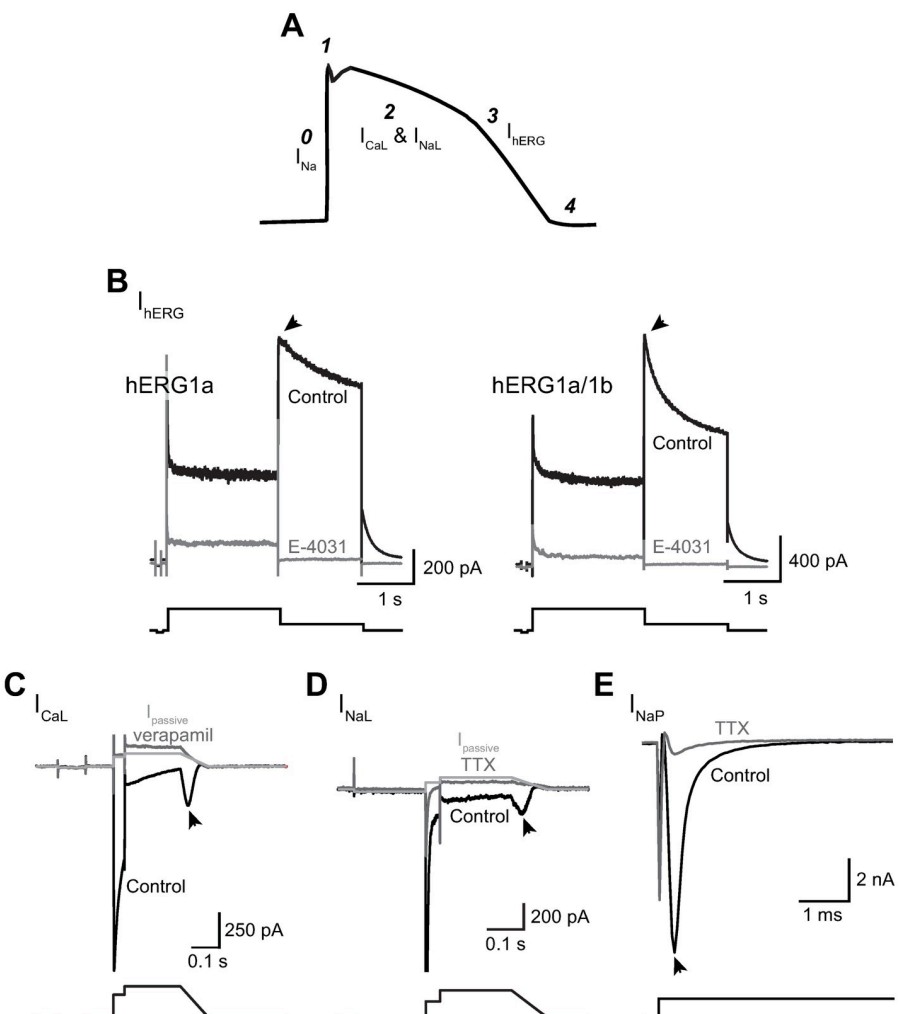

**Fig 1. Ionic currents studied and voltage protocols used to evoke them. A)** Schematic diagram of a ventricular AP, with phases of the AP indicated by numbers and ionic currents contributing to these phases labelled. **B-E)** Representative current traces (upper panels) recorded from cells overexpressing the specified ion channel proteins and voltage protocols used to evoke them (lower panels). Control traces are shown in black; following bath application of a selective blocker, dark gray is used. For $I_{CaL}$ recordings in **(C)** and $I_{NaL}$ recordings in **(D)**, $I_{passive}$ traces in light gray were calculated using input resistance values derived from the control traces (see "Methods"). $I_{NaL}$ and $I_{NaP}$ were studied using the same voltage protocol. $I_{NaL}$ was recorded in the presence of 150 nM ATX-II and its amplitude quantified as the peak inward current during the voltage ramp down phase. $I_{NaP}$ was recorded in the absence of ATX-II and quantified as the peak inward current during the -15 mV step. The selective blocker used to inhibit $I_{hERG}$ was 1 μM E-4031; $I_{CaL}$, 100 μM verapamil; and $I_{NaL}$ and $I_{NaP}$, 30 μM TTX. The arrowhead in each panel denotes the region where each current was measured to quantify drug effects.

Fig 1B shows outward currents recorded from two representative cells expressing hERG1a and hERG1a/1b subunits, in control solution (black traces) and following bath application of the hERG channel blocker E-4031 (1 μM; dark gray traces). The tail current during the repolarization step to -50 mV was nearly abolished by E-4031 in both hERG1a and hERG1a/1b cells, indicating negligible background or leak current at this membrane voltage (tail current amplitude in control solution: 1640.8 ± 92.9 pA; following E-4031 application: 29.0 ± 2.0 pA; E-4031-insensitive current: 2.0 ± 0.2%; n = 81). Therefore, drug effect on $I_{hERG}$ was quantified using changes in the tail current amplitude. Note that deactivation of the tail current in

hERG1a/1b cells was faster than that in hERG1a cells (see summary spreadsheet for $I_{hERG}$ experiments at https://osf.io/tjuev/), consistent with different gating behavior for homomeric and heteromeric hERG channels [21].

Fig 1C shows current traces recorded from a representative cell expressing $Ca_V1.2/\beta_2/\alpha_2\delta1$ subunits, in control solution (black trace) and following bath application of the $Ca_V1.2$ channel blocker verapamil (100 μM; dark gray trace). The light gray trace is the calculated passive current ($I_{passive}$) for the control current trace. Despite mismatch at the +30 mV step, in the majority of the cells $I_{passive}$ and verapamil-insensitive current traces aligned well at the ramp-down voltage phase where the $I_{CaL}$ amplitude was quantified. Therefore, in this study $I_{CaL}$ amplitude was quantified by subtracting $I_{passive}$. For a subset of cells, $I_{passive}$ and residual current following complete $I_{CaL}$ inhibition did not align well at the voltage ramp-down phase, suggesting the presence of an endogenous current. For these cells the residual current trace was used for current subtraction to estimate $I_{CaL}$ amplitude.

Fig 1D and 1E show current traces recorded from representative cells overexpressing $Na_V1.5\alpha/\beta1$ subunits, in the presence of 150 nM ATX-II to study $I_{NaL}$ (Fig 1D) and without to study $I_{NaP}$ (Fig 1E). Current trace recorded in control solution is shown in black; following 30 μM TTX application, in dark gray. In Fig 1D, $I_{passive}$ was calculated using the control trace and is shown in light gray. Note that the TTX-insensitive current trace closely matched $I_{passive}$. Therefore, in this study $I_{NaL}$ was measured as the peak inward current at the ramp-down voltage phase and was quantified by subtracting $I_{passive}$. $I_{NaP}$ was not $I_{passive}$-subtracted given its large amplitude (in control solution: -7578.4 ± 865.4 pA; in TTX solution: -888.3 ± 148.6 pA; TTX-insensitive current: 10%; n = 25).

Fig 2 shows the time course plots of representative pharmacology experiments for $I_{hERG}$ (Fig 2A), $I_{CaL}$ (Fig 2B), $I_{NaL}$ (Fig 2C), and $I_{NaP}$ (Fig 2D).

**Multi-channel pharmacology for buprenorphine.** Fig 3 shows the concentration-inhibition plots for buprenorphine against $I_{hERG}$ (Fig 3A), $I_{CaL}$ (Fig 3B), $I_{NaL}$ (Fig 3C), and $I_{NaP}$ (Fig 3D). The $IC_{50}$ and $n_H$ values are summarized in Table 1. Buprenorphine's effects on homomeric and heteromeric hERG channels are not statistically different.

Fig 3E shows the overlay of all concentration-inhibition plots and buprenorphine's free $C_{max}$ (0.00087 μM). There is no overlap between these *in vitro* and clinical concentrations. The ratios of $IC_{50}$s for individual ionic currents and buprenorphine's free $C_{max}$ are presented in Table 1. This ratio for hERG1a, considered as the "safety margin" by drug developers, is 10,046. For reference, a safety margin of 30 to 45 and above has been proposed by several retrospective studies to indicate a low likelihood of drug-induced $QT_C$ prolongation [28–30].

**Norbuprenorphine.** Fig 4 shows the concentration-inhibition plots for norbuprenorphine against $I_{hERG}$ (Fig 4A), $I_{CaL}$ (Fig 4B), $I_{NaL}$ (Fig 4C), and $I_{NaP}$ (Fig 4D). Fig 4E shows the overlay of all concentration-inhibition plots and norbuprenorphine's total $C_{max}$ (0.22 μM). There is no overlap between these *in vitro* and clinical concentrations. Table 1 summarizes the values of $IC_{50}$, $n_H$, and the ratios of $IC_{50}$s and total $C_{max}$ for norbuprenorphine. The safety margin of this drug, calculated using total and not free $C_{max}$ as the percent of protein binding has not been empirically determined, is 1,634.

**Methadone.** Racemic methadone, also known as *R,S*-methadone or (±)-methadone, was used in this study since this is the formulation approved in the United States to manage opioid addiction. Fig 5 shows the concentration-inhibition plots of methadone on $I_{hERG}$ (Fig 5A), $I_{CaL}$ (Fig 5B), $I_{NaL}$ (Fig 5C), and $I_{NaP}$ (Fig 5D). Fig 5E shows the overlay of all concentration-inhibition plots and methadone's free $C_{max}$ (0.44 μM). Unlike buprenorphine and norbuprenorphine, there is clear overlap between the *in vitro* and clinical concentrations for methadone. Table 1 summarizes the values of $IC_{50}$, $n_H$, and the ratios of $IC_{50}$s and free $C_{max}$ for

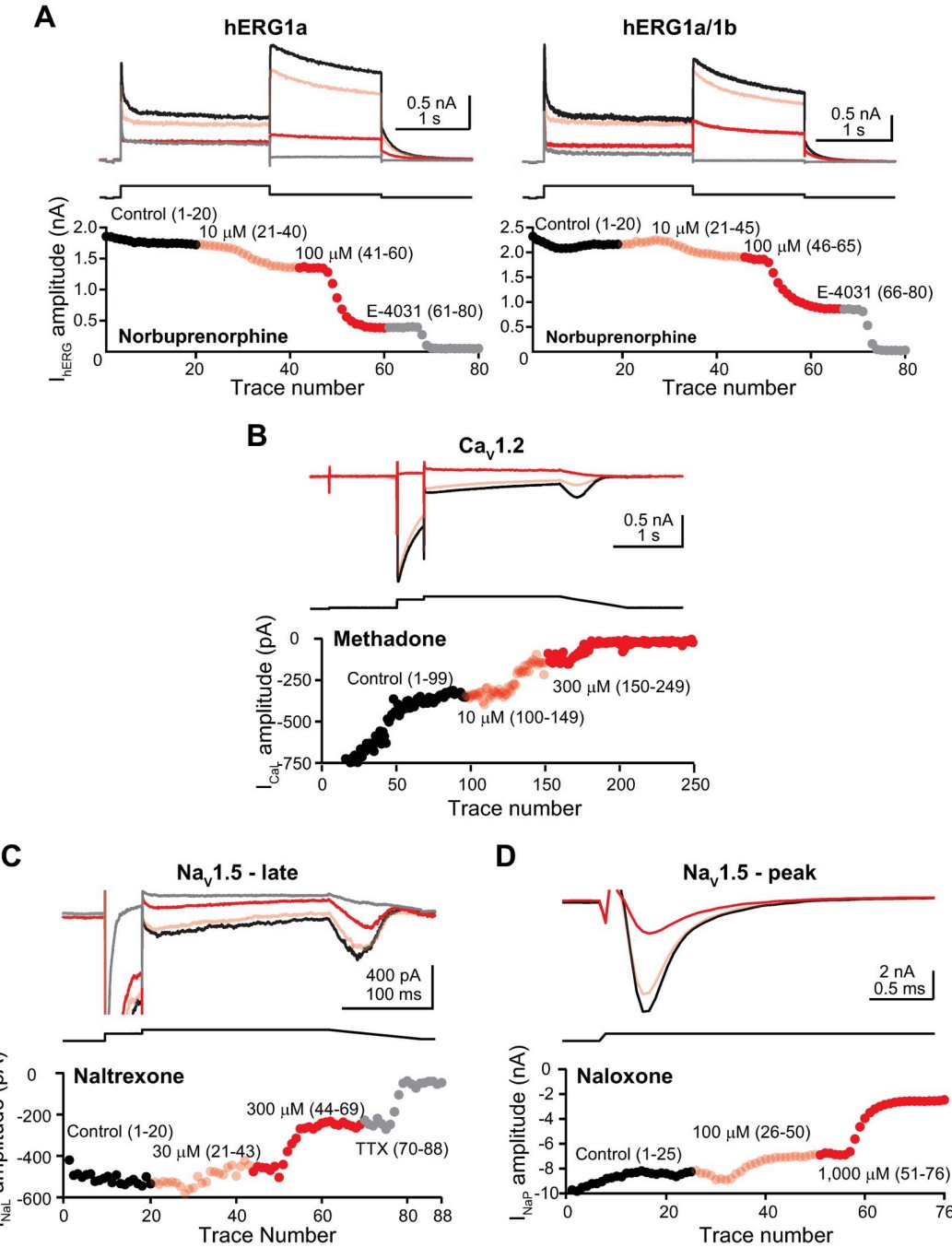

**Fig 2. Time course plots of representative pharmacology experiments. A)** *Upper panels*, representative $I_{hERG}$ traces recorded from a hERG1a- (left panel) and a hERG1a/1b-expressing cell (right panel) in control solution, followed by sequential applications of 10 μM and 100 μM norbuprenorphine and then 1 μM E-4031. *Lower panels*, time course plots of the $I_{hERG}$ amplitude for the duration of these experiments. The current traces and time course plot data points are color-coded according to drug(s) and concentrations tested, and the numbers in the parentheses indicate current trace numbers recorded under the specified test condition. Panels **(B)** to **(D)** are organized similarly. **B)** $I_{CaL}$ recorded in control solution and followed by sequential applications of 10 μM and 300 μM methadone. Note that due to pronounced activity-dependent rundown typical of this current at the beginning of the experiment, data points for the first 14 current traces were off the Y-axis scale used for illustration. **C)** $I_{NaL}$ recorded in control solution and followed by sequential applications of 30 μM and 300 μM naltrexone and then 30 μM TTX. **D)** $I_{NaP}$ recorded in control solution and followed by sequential applications of 100 μM and 1000 μM naloxone.

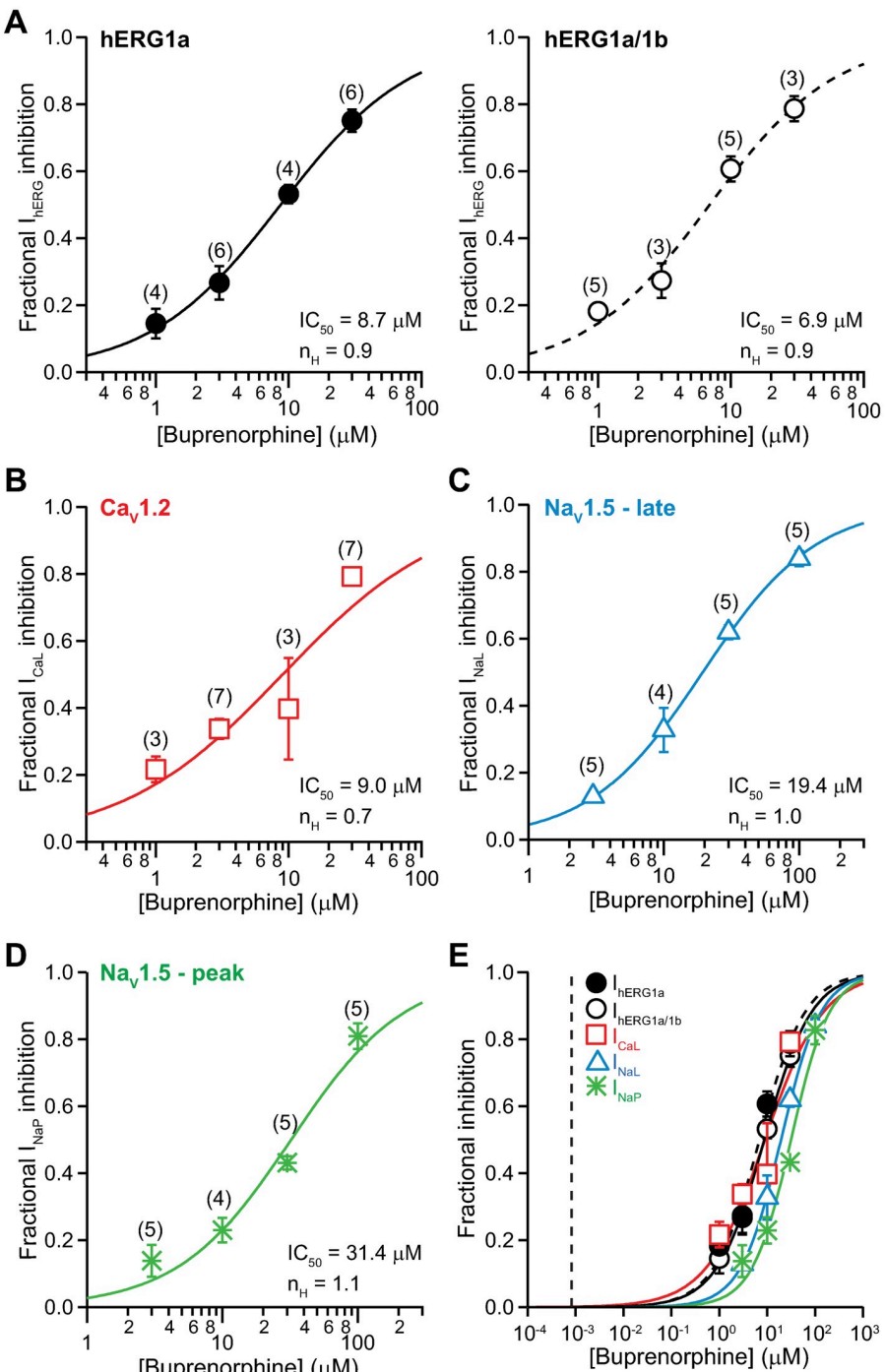

**Fig 3. Effects of buprenorphine.** Concentration-inhibition plots of buprenorphine on $I_{hERG}$ (**A**), $I_{CaL}$ (**B**), $I_{NaL}$ (**C**), and $I_{NaP}$ (**D**). Data are shown as mean ± SEM. Each plot was fit with the Hill equation to the mean values, with maximum and minimum constrained to 1 and 0, respectively. The numbers in the parentheses correspond to the number of data points obtained for each drug concentration. **E)** An overlay of concentration-inhibition plots of buprenorphine on all tested ionic currents. The vertical dashed line indicates free $C_{max}$ of 0.00087 μM.

**Table 1. Potencies of buprenorphine, norbuprenorphine, methadone, naltrexone, and naloxone on inhibiting $I_{hERG}$, $I_{CaL}$, $I_{NaL}$, and $I_{NaP}$.**

| | $C_{max}$ (in μM; *free; #total) | Ionic current | $IC_{50}$ (μM) at 37°C from this study (mean ± SD) | $n_H$ (mean ± SD) | $IC_{50}$/free or total $C_{max}$ | Literature $IC_{50}$ (μM) (^room temperature; &37°C; +ambient) |
|---|---|---|---|---|---|---|
| **Buprenorphine** | 0.00087* | hERG1a | 8.7 ± 0.4 | 0.9 ± 0.03 | 10046.1 | 7.5^ [10] |
| | | hERG1a/1b | 6.9 ± 0.9 | 0.9 ± 0.1 | 7967.7 | |
| | | $I_{CaL}$ | 9.0 ± 3.5 | 0.7 ± 0.2 | 10344.8 | |
| | | $I_{NaL}$ | 19.4 ± 0.5 | 1.0 ± 0.3 | 22401.8 | |
| | | $I_{NaP}$ | 31.4 ± 5.7 | 1.1 ± 0.2 | 36258.7 | 55^ [11] |
| **Norbuprenorphine** | 0.022# | hERG1a | 36.6 ± 3.7 | 1.2 ± 0.1 | 1633.9 | |
| | | hERG1a/1b | 30.9 ± 5.0 | 0.9 ± 0.1 | 1379.5 | |
| | | $I_{CaL}$ | 27.8 ± 5.4 | 0.8 ± 0.1 | 1327.3 | |
| | | $I_{NaL}$ | 4.0 ± 0.5 | 1.1 ± 0.1 | 178.6 | |
| | | $I_{NaP}$ | 8.0 ± 0.7 | 1.0 ± 0.1 | 357.1 | |
| **Methadone** | 0.44* | hERG1a | 2.8 ± 0.3 | 0.9 ± 0.1 | 6.4 | 4.8^ [46] |
| | | | | | | 9.8^ [10] |
| | | | | | | 19^ [47] |
| | | | | | | 3& [47] |
| | | | | | | 1.7+ [48] |
| | | hERG1a/1b | 2.7 ± 0.9 | 0.9 ± 0.1 | 6.2 | |
| | | $I_{CaL}$ | 5.5 ± 1.1 | 1.1 ± 0.2 | 12.6 | phasic 26.3; tonic 7.7+ [48] |
| | | $I_{NaL}$ | 8.5 ± 0.6 | 0.9 ± 0.1 | 19.4 | 62^ [49] |
| | | $I_{NaP}$ | 40.1 ± 3.6 | 1.3 ± 0.1 | 91.6 | 90^ [49] |
| | | | | | | Phasic 11.2; tonic 5.5+ [48] |
| **Naltrexone** | 0.028* | hERG1a | 31.1 ± 5.4 | 0.8 ± 0.1 | 1108.9 | |
| | | hERG1a/1b | 32.3 ± 8.6 | 0.7 ± 0.1 | 1151.7 | |
| | | $I_{CaL}$ | 1067.7 ± 756.0 | 0.6 ± 0.2 | 38132.1 | |
| | | $I_{NaL}$ | 148.5 ± 15.6 | 0.8 ± 0.1 | 5295.1 | |
| | | $I_{NaP}$ | 139.5 ± 24.5 | 0.8 ± 0.1 | 4974.1 | |
| **Naloxone** | 0.013* | hERG1a | 37.3 ± 4.0 | 0.9 ± 0.1 | 2878.1 | 74.3^ [45] |
| | | hERG1a/1b | 33.9 ± 2.8 | 0.9 ± 0.1 | 2615.7 | |
| | | $I_{CaL}$ | 392.4 ± 71.7 | 0.4 ± 0.03 | 30277.8 | |
| | | $I_{NaL}$ | 284.7 ± 39.3 | 1.0 ± 0.2 | 21967.6 | |
| | | $I_{NaP}$ | 451.8 ± 13.9 | 0.9 ± 0.02 | 34861.1 | |

methadone. The safety margin of methadone is 6.4. At free $C_{max}$, methadone is estimated to inhibit $I_{hERG}$ by 16%, and $I_{CaL}$ and $I_{NaL}$ by 6%.

**Naltrexone.** Fig 6 shows the concentration-inhibition plots of naltrexone on $I_{hERG}$ (Fig 6A), $I_{CaL}$ (Fig 6B), $I_{NaL}$ (Fig 6C), and $I_{NaP}$ (Fig 6D). The overlay of all concentration-inhibition plots and naltrexone's free $C_{max}$ (0.028 μM) shows no overlap between the *in vitro* and clinical concentrations (Fig 6E). Table 1 shows the values of $IC_{50}$, $n_H$, and the ratios of $IC_{50}$s and free $C_{max}$ for naltrexone. The safety margin of this drug is 1,109.

**Naloxone.** Fig 7 shows the concentration-inhibition plots of naloxone on $I_{hERG}$ (Fig 7A), $I_{CaL}$ (Fig 7B), $I_{NaL}$ (Fig 7C), and $I_{NaP}$ (Fig 7D). Fig 7E shows the overlay of all concentration-inhibition plots and naloxone's free $C_{max}$ (0.013 μM). Table 1 shows the values of $IC_{50}$, $n_H$, and the ratios of $IC_{50}$s and total $C_{max}$ for naloxone. The safety margin of this drug is 2,878.

**iPSC-CMs.** Drug effects on cardiac APs were tested using iPSC-CMs and MEA platform. Fig 8 shows concentration-dependent changes in the field potential duration (FPD), beat period (BP), and spike amplitude (SA) of iPSC-CMs treated with these drugs as well as each

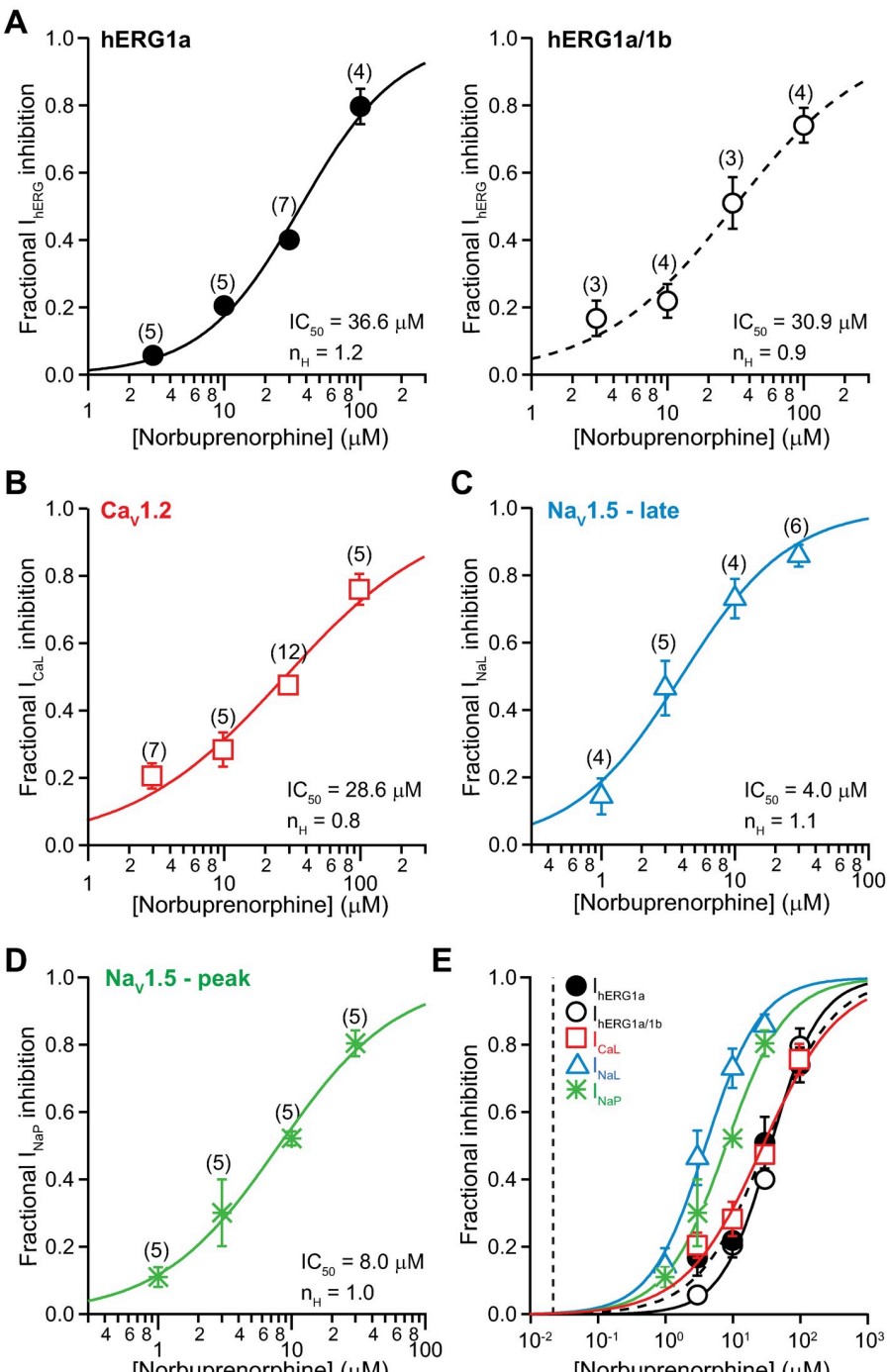

**Fig 4. Effects of norbuprenorphine.** Concentration-inhibition plots of norbuprenorphine on $I_{hERG}$ **(A)**, $I_{CaL}$ **(B)**, $I_{NaL}$ **(C)**, and $I_{NaP}$ **(D)**. **E)** An overlay of all concentration-inhibition curves. The vertical dashed line indicates total $C_{max}$ of 0.022 µM.

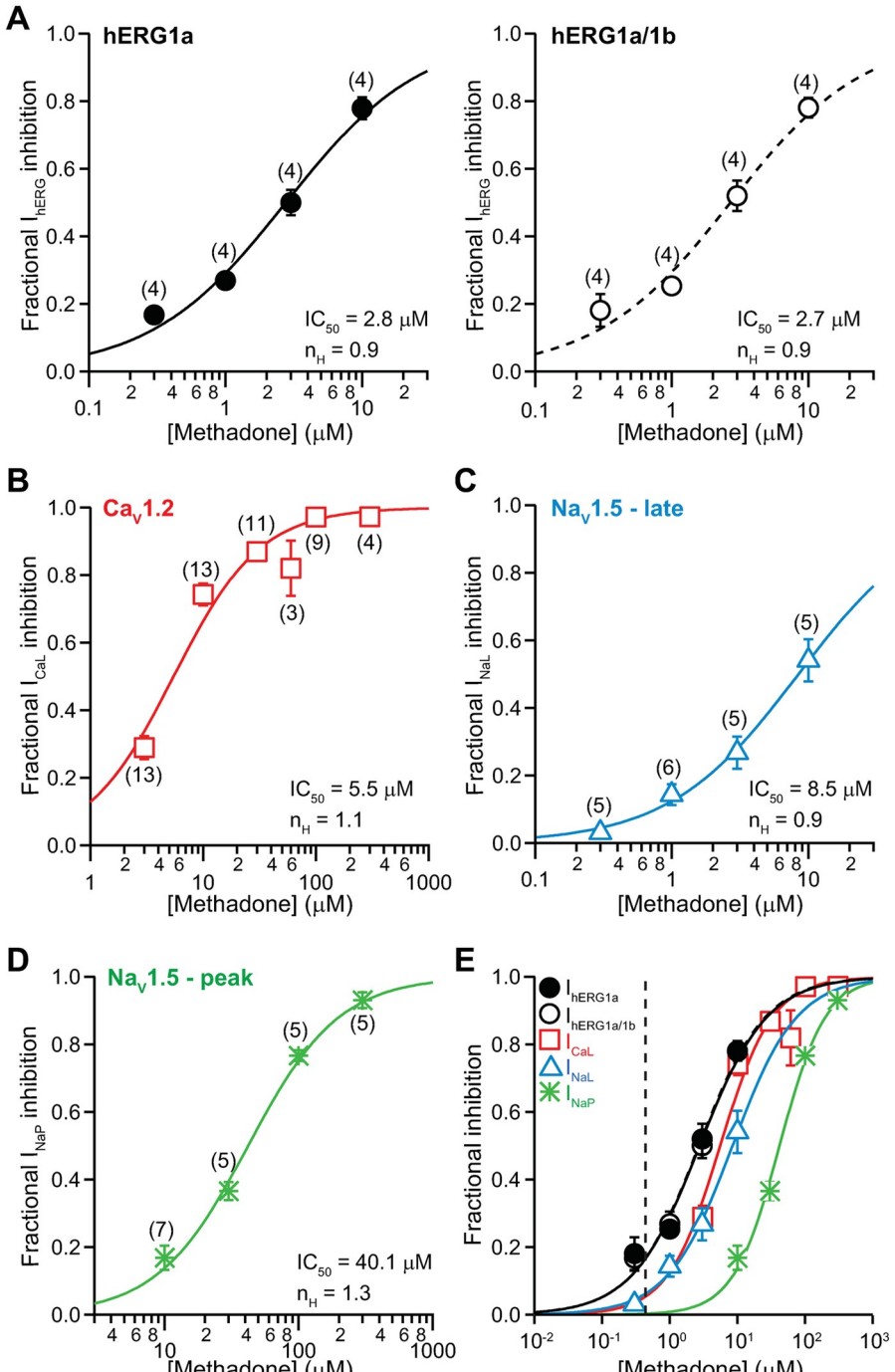

**Fig 5. Effects of methadone.** Concentration-inhibition plots of methadone on $I_{hERG}$ **(A)**, $I_{CaL}$ **(B)**, $I_{NaL}$ **(C)**, and $I_{NaP}$ **(D)**. **E)** An overlay of concentration-inhibition curves for all studied currents. The vertical dashed line indicates free $C_{max}$ of 0.44 μM.

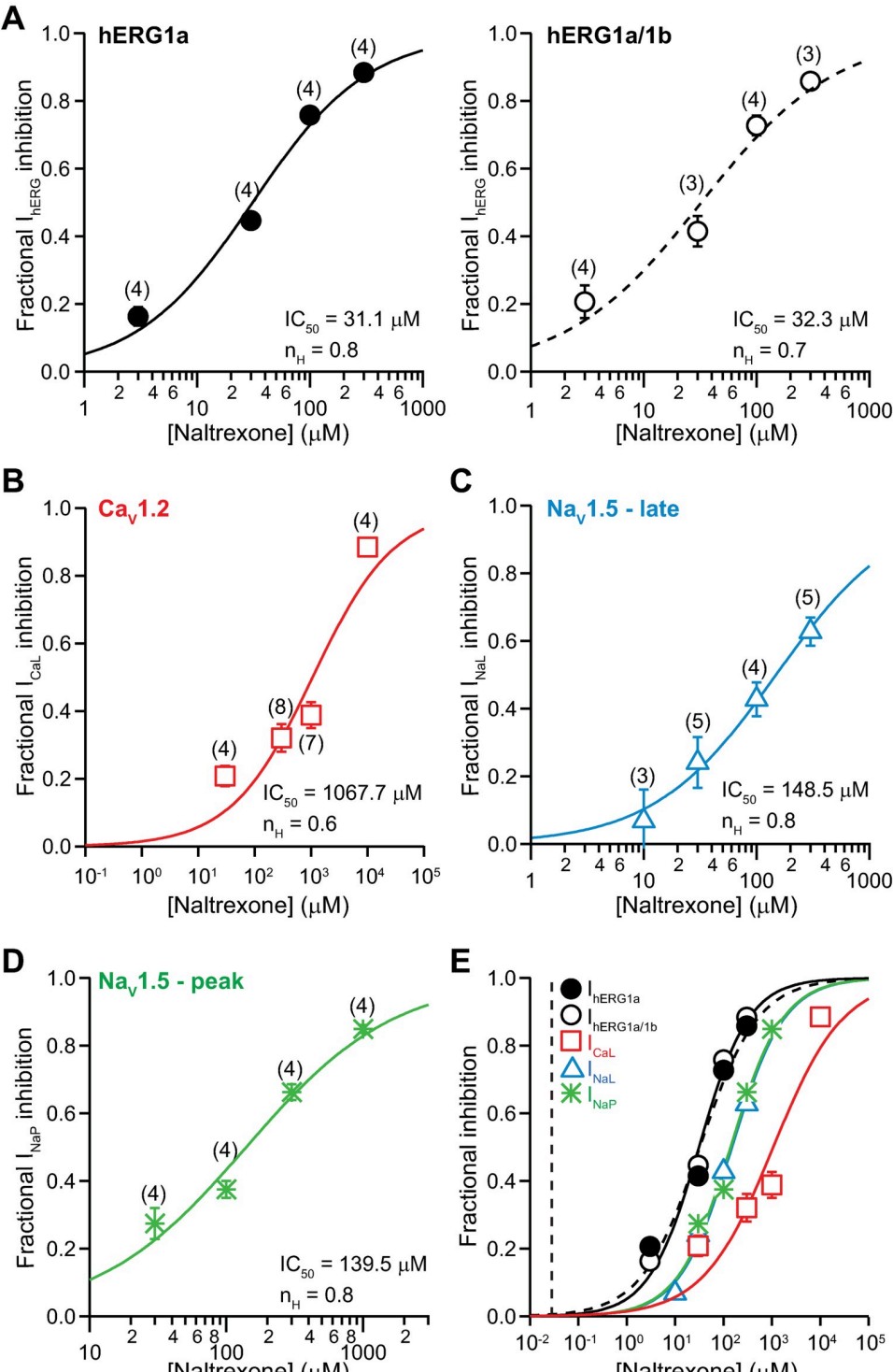

**Fig 6. Effects of naltrexone.** Concentration-inhibition plots of naltrexone on $I_{hERG}$ (**A**), $I_{CaL}$ (**B**), $I_{NaL}$ (**C**), and $I_{NaP}$ (**D**). **E)** An overlay of concentration-inhibition curves for all studied currents. The vertical dashed line indicates free $C_{max}$ of 0.028 µM.

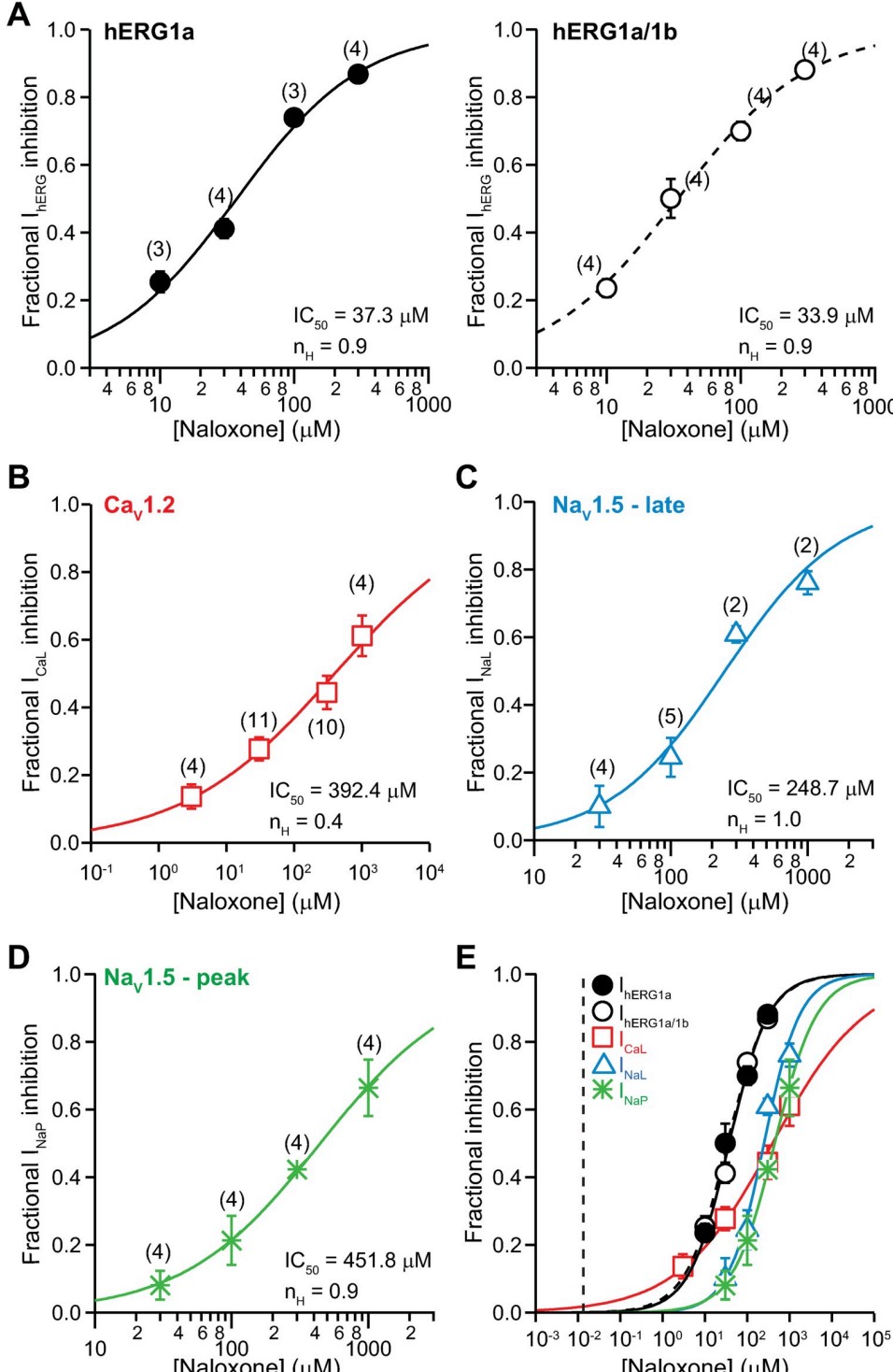

**Fig 7. Effects of naloxone.** Concentration-inhibition plots of naloxone on $I_{hERG}$ (**A**), $I_{CaL}$ (**B**), $I_{NaL}$ (**C**), and $I_{NaP}$ (**D**). **E)** An overlay of concentration-inhibition curves for all studied currents. The vertical dashed line indicates free $C_{max}$ of 0.013 μM.

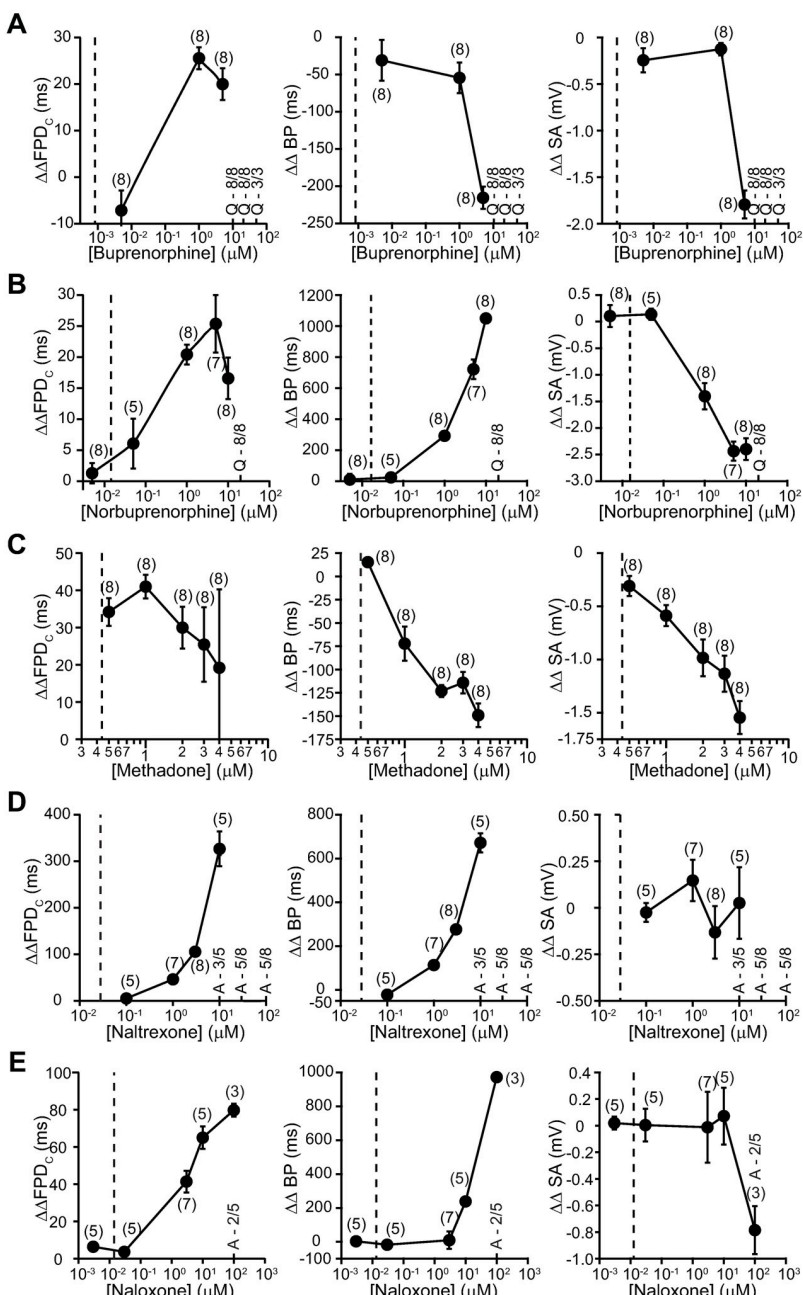

**Fig 8. Field recordings of iPSC-CMs.** Concentration-dependent changes in $\Delta\Delta FPD_C$, $\Delta\Delta BP$, and $\Delta\Delta SA$ of iPSC-CMs treated with various concentrations of buprenorphine **(A)**, norbuprenorphine **(B)**, methadone **(C)**, naltrexone **(D)**, and naloxone **(E)** on MEA plates. The effect of opioids was analyzed 30 minutes after adding respective drugs to the wells. 'Quiescence' was observed in few instances at higher drug concentrations. These events are denoted as "Q" in panels along with the number of replicate wells that showed 'quiescence'. Arrythmia-like events were also observed in iPSC-CMs when treated with naltrexone at 10, 30 and 100 μM and naloxone at 100 μM. These events are denoted as "A" in panels along with the number of replicates. Data are represented as mean ± SEM. The vertical dashed line indicates free $C_{max}$ for buprenorphine, methadone, naltrexone, and naloxone, and total $C_{max}$ for norbuprenorphine.

drug's free or total $C_{max}$. Note that in these plots, the lines connecting the summary datapoints were for illustrative purposes and do not describe the relationship of how measured parameters change with increasing drug concentrations.

At 1 and 3 μM, buprenorphine caused increases in ΔΔFPD in spontaneously beating iPSC-CMs. For reference, 1 μM is 1,111X higher than buprenorphine's free $C_{max}$. Buprenorphine also caused reductions in ΔΔBP at 1 and 3 μM and ΔΔSA at 3 μM. At 10 μM and above, buprenorphine induced quiescence in all cells.

Fig 8B shows the effects of norbuprenorphine. This drug also caused concentration-dependent increases in ΔΔFPD at 0.05–10 μM and ΔΔBP at 1–10 μM, and decreases in ΔΔSA at 0.05–10 μM. For reference, norbuprenorphine's total $C_{max}$ is 0.022 μM, and simulation predicted this metabolite to be extensively plasma protein-bound as the parent molecule (95 ± 5%; see "Methods"). Thus, the *in vitro* concentrations tested in these studies are likely much higher than the free drug level achieved with therapeutic dosing.

Fig 8C shows the effects of methadone. The lowest concentration tested was 0.5 μM and approximately corresponds to free $C_{max}$. Significant increase in ΔΔFPD is evident at this concentration. Methadone also caused concentration-dependent reductions in ΔΔBP and ΔΔSA, consistent with having effects on multiple cardiac ionic currents. These changes emerged at 1 μM (2.3X free $C_{max}$) and 0.5 μM (1.1X free $C_{max}$), respectively.

Fig 8D and 8E show the effects of naltrexone and naloxone, respectively. Both drugs caused concentration-dependent increases in ΔΔFPD and ΔΔBP. For naltrexone, these electrophysiological changes emerged at 1 μM (28.6X total $C_{max}$); and for naloxone, at 3 μM (125X free $C_{max}$) for ΔΔFPD and 10 uM for ΔΔBP (416.7X free $C_{max}$). No effect on ΔΔSA was observed for either drug up to 10 μM. Naltrexone at 10, 30, and 100 μM and naloxone at 100 μM induced arrythmia-like events in 3–5 out of 8 wells and 2 out of 5 wells, respectively.

## Discussion and conclusions

Buprenorphine and methadone are opioid receptor agonists that have been associated with $QT_C$ prolongation, with methadone generally accepted as having high TdP risk and buprenorphine having no documentation of associated TdP in the medical literature [1, 26, 31–34]. This study shows that at clinically relevant concentrations, buprenorphine and its metabolite norbuprenorphine do not inhibit inward ($I_{CaL}$, $I_{NaL}$, and $I_{NaP}$) or outward ($I_{hERG}$) cardiac ionic currents (Fig 1) or alter AP characteristics in iPSC-CMs (Figs 3, 4, 8A and 8B). In contrast, methadone inhibits $I_{hERG}$, $I_{CaL}$, and $I_{NaL}$, and increases FPD in iPSC-CMs (Figs 5 and 8C). The mechanism of $QT_C$ prolongation is thus opioid moiety-specific: for buprenorphine, this mechanism remains unresolved; for methadone, hERG channel block plays a role. Naltrexone and naloxone are two non-specific opioid receptor antagonists used to treat opioid use disorder and as emergency treatment for opioid overdose, respectively. Neither naltrexone nor naloxone has acute effects on the tested ionic currents or iPSC-CMs at clinically relevant concentrations (Figs 6, 7, 8D and 8E). In the BUTRANS® study, buprenorphine was not associated $QT_C$ prolongation when naltrexone was co-administered [1]. In light of the present results, this clinical observation suggests that $QT_C$ prolongation associated with buprenorphine is initiated by opioid receptor activation.

Buprenorphine has mixed effects on different opioid receptor subtypes. It is a partial agonist for MOR, an agonist for nociception or OR-like 1 (ORL1) receptor, and an antagonist at delta opioid receptor (DOR) and kappa opioid receptor (KOR) [35]. Table 2 shows the inhibition constants ($K_i$) or binding affinities of buprenorphine and other tested drugs on different human opioid receptor subtypes. In sheep Purkinje fibers and rat papillary muscles, buprenorphine at 1–10 μM reduced AP height and maximum rate of rise ($V_{max}$) while it prolonged

**Table 2. Binding affinity at human opioid receptors.**

| | $K_i$ (nM) | | | | |
|---|---|---|---|---|---|
| **Drug** | **μ** | **δ** | **κ** | **ORL1** | **Reference** |
| Buprenorphine | 0.0027 | 33 | 0.0021 | 25,000 | [60] |
| Steady state free $C_{max}$ = 0.87 nM | 0.52 | | | | [61] |
| | 0.08 | 0.42 | 0.11 | 285 | [37] |
| | 1.5 | 6.1 | 2.5 | 77.4 | [62] |
| | 1.5 | 4.5 | 0.8 | | [63] |
| | 0.21 | | | | [64] |
| | 0.21 | 2.9 | 0.62 | | [65] |
| | 1.8 | | | | [66] |
| | 1.5 | 4.5 | 0.8 | | [67] |
| Norbuprenorphine | 0.0018 | 1300 | 0.0013 | | [60] |
| Steady state total $C_{max}$ = 22 nM | 0.07 | 3.17 | 0.91 | 7330 | [37] |
| Methadone | 14 | | | | [61] |
| Free $C_{max}$ = 438 nM | 3.7 | | | | [68] |
| | 5.6 | | | | [69] |
| | 0.6 | 132.2 | 323.5 | | [63] |
| | 3.37 | | | | [37] |
| | 7.7 | | | | [66] |
| | 3.38 | | | | [64] |
| Naltrexone | 1.1 | | | | [61] |
| Free $C_{max}$ = 28 nM | 0.26 | 117 | 5.15 | | [70] |
| | 0.2 | 10.8 | 4 | | [63] |
| | 0.5 | | 1 | | [71] |
| Naloxone | 3.3 | | | | [61] |
| Free $C_{max}$ = 13 nM | 0.8 | | | | [68] |
| | 0.79 | 76 | 1.1 | | [72] |
| | 2.4 | | | | [73] |
| | 1.4 | 67.5 | 2.5 | | [63] |
| | 3 | | | | [74] |
| | 1.51 | | | | [64] |
| | 7 | | 4 | | [71] |
| | 0.66 | 120 | 1.2 | | [65] |
| | 0.6 | | | | [66] |

APD [36]. These electrophysiological changes suggest inhibition of $I_{NaP}$ and $I_{hERG}$, as demonstrated in subsequent patch clamp studies [10, 11] (see Table 1 for $IC_{50}$s). The present study confirms and extends the previous work, demonstrating that buprenorphine inhibits $I_{hERG}$, $I_{NaP}$, and additionally $I_{CaL}$ and $I_{NaL}$, with $IC_{50}$s in the micromolar to tens of millimolar concentration range (Fig 3A–3D; Table 1). Given the large difference between the *in vitro* concentrations and buprenorphine's free $C_{max}$ (Fig 3E), direct inhibition of these ionic currents seems unlikely following therapeutic dosing. Results from iPSC-CMs similarly show a lack of drug effect on repolarization at clinically relevant concentrations (Fig 8A). Together, these data demonstrate that the direct cellular targets of buprenorphine that mediate $QT_C$ prolongation are not hERG, $Ca^{2+}$, or $Na^+$ channels or present on iPSC-CMs.

$QT_C$ prolongation by buprenorphine could also be mediated by its metabolite norbuprenorphine, which is a partial agonist for MOR and KOR, and a full agonist for DOR and ORL1

[37] (Table 2). Norbuprenorphine's $IC_{50}$s against the tested ionic currents are in the micromolar to tens of micromolar concentration range, values that far exceed the drug's total $C_{max}$ following therapeutic dosing (Fig 4E; Table 1). Experiments on iPSC-CMs show that at 0.005 μM (0.23X total $C_{max}$), norbuprenorphine caused no change in ΔΔFPD$_C$, ΔΔBP, or ΔΔSA (Fig 8B). At 0.05 μM and above (>2.3X total $C_{max}$), norbuprenorphine caused concentration-dependent increases in ΔΔFPD$_C$ and ΔΔBP, and decrease in ΔΔSA. That ΔΔFPD$_C$ began to increase at 0.05 μM was unexpected based on the ion channel data presented in Fig 4. Nonetheless, this concentration is likely much higher than the free drug level after therapeutic dosing, as norbuprenorphine is predicted to exhibit extensive plasma protein binding as buprenorphine. Thus, effects of norbuprenorphine on cardiac ion channels or iPSC-CMs cannot explain QT$_C$ prolongation associated with buprenorphine.

Naltrexone and its active metabolite 6β-naltrexol are competitive antagonists at MOR and KOR, and to a lesser extent at DOR [38] (Table 2). Naltrexone inhibited the tested ionic currents with $IC_{50}$s in the tens of micromolar to millimolar concentration range (Fig 6A–6D), values far higher than the drug's free $C_{max}$ following therapeutic dosing (Fig 6E; Table 1). Results from iPSC-CMs similarly showed a lack of drug effect on APs at clinically relevant concentrations (Fig 8D). These results suggest that naltrexone does not reverse or normalize QT$_C$ prolongation by buprenorphine by acting on hERG, Ca$^{2+}$, or Na$^+$ channels or cellular targets intrinsic to iPSC-CMs. Rather, this drug likely prevents QT$_C$ prolongation by preventing buprenorphine from activating opioid receptors.

Naloxone is a competitive antagonist at MOR and KOR, and a weak antagonist at DOR [39] (Table 2). Direct electrophysiological effects of this drug on cardiomyocytes and cardiac ion channels have been reported in many studies. In the micromolar to sub-millimolar concentration range, naloxone has been shown to prolong ventricular APD [40–44] and decrease V$_{max}$ [41, 42]—changes suggestive of drug effects on multiple cardiac ion channels. Indeed, in the tens of micromolar concentration range, naloxone has been found to decrease I$_{Na}$, I$_{to}$, I$_{CaL}$, and I$_{Kr}$, and increase I$_{K1}$ (inward rectifier current that sets the resting membrane potential) in native myocytes [44], and inhibit I$_{hERG}$ in overexpression cells [45] (Table 1). The present data are consistent with the literature, showing that naloxone inhibits I$_{hERG}$, I$_{CaL}$, I$_{NaL}$, and I$_{NaP}$ with $IC_{50}$s ranging from tens to hundreds of micromolar (Fig 6; Table 1). Given the drug's free $C_{max}$ of 0.013 μM, however, direct inhibition of the aforementioned cardiac ionic currents from past or present studies seems unlikely following therapeutic dosing. Consistent with this interpretation, naloxone at clinically relevant concentrations produced no change in ΔΔFPD$_C$, ΔΔBP, or ΔΔSA in iPSC-CMs (Fig 8E).

Methadone is used as treatment of opioid use disorder and treatment for pain. While the literature states that this drug is a full agonist at MOR, following therapeutic dosing methadone can reach a level that binds to DOR and KOR as well (Table 2). Methadone is associated with QT$_C$ prolongation and TdP [26], and was studied here to provide a comparison for buprenorphine. Past studies have found methadone to be a multi-ion channel blocker, with $IC_{50}$s for I$_{hERG}$ [10, 46–48], I$_{CaL}$ [48], I$_{NaL}$ [49], and I$_{NaP}$ [48, 49] all in the micromolar to tens of micromolar range. The present results are largely consistent with the literature (Fig 5; Table 1), showing that methadone inhibits I$_{hERG}$ by 16% and I$_{CaL}$ and I$_{NaL}$ by 6% each at the drug's free $C_{max}$. The overall consequence of inward and outward current inhibition is increased ΔΔFPD$_C$ and ΔΔBP in iPSC-CMs at 0.5 μM (1.1X free $C_{max}$) (Fig 8C). Similar findings of repolarization delay by methadone have also been found in a study of sheep Purkinje fibers [50] and human stem cell-derived cardiomyocytes [48]. Thus, following therapeutic dosing, repolarization delay by methadone that results from acute hERG channel block can be expected.

An important limitation of this study is the lack of concentration analysis to confirm actual drug concentrations exposed to the recorded cells. Actual concentrations may differ from

nominal drug concentrations due to issues related to stability, solubility, adsorption in the perfusion apparatus, and in the case of MEA experiments, the presence of serum proteins in the medium [51–53]. Such possibility has led to the question whether the central conclusion of this study–that buprenorphine and norbuprenorphine do not directly block hERG channels with therapeutic dosing–can still be supported. Two lines of reasoning suggest "yes". First, despite the uncertainty in *in vitro* drug concentrations, several retrospective studies have pointed to a hERG safety margin of 30 to 45 and above to identify drugs with low QT prolongation risk due to direct hERG channel block. Buprenorphine's safety margin is far greater than that threshold and excess of 10,000. Even norbuprenorphine's safety margin, determined using total rather than free $C_{max}$, is excess of 1,600. Second, a previous study that compared drug loss for 39 compounds showed the most extensive loss to be 74% [51]. Even with that level of drug loss, buprenorphine's safety margin would still be 2600, whereas that for norbuprenorphine determined using total $C_{max}$ would still be 416. Therefore, the lack of concentration analysis in this study may affect the actual safety margin values for buprenorphine and norbuprenorphine. However, due to the magnitude of these values, the study conclusion— that acute and direct block of hERG channels by buprenorphine (and norbuprenorphine) is unlikely to occur with therapeutic dosing and cannot explain QT prolongation seen in Harris et al. (2017) [1]–is not altered.

Opioid receptor agonists have been shown to affect the release of neurotransmitters, including acetylcholine due to vagal stimulation [54, 55] and noradrenaline due to sympathetic nerve stimulation [56, 57] to the heart in basic science studies. Thus, it seems plausible that buprenorphine-associated $QT_C$ prolongation is mediated by activation of presynaptic opioid receptors that alter neuromodulatory tone to the heart, resulting in repolarization delay. In terms of opioid receptor localization in the human heart, one study has reported diffuse immunoreactivities for MOR, DOR, and KOR in ventricular myocardial cells [58]. These results seem to be at odds with the findings that buprenorphine does not alter electrical behavior of animal ventricular myocytes [36] or human iPSC-CMs (Fig 8A and 8B) at clinically relevant concentrations of nanomolar range. Additional studies with different opioid receptor antibodies to confirm receptor localization on human primary ventricular myocytes will be helpful to reconcile these discrepancies.

Sobanski and colleagues also reported strong immunolabeling for MOR, DOR, and KOR on nerve fibers in human ventricular tissues, with MOR and DOR immunoreactivities on sparse individual nerve fibers, and KOR immunoreactivity predominantly on intrinsic cardiac adrenergic (ICA) cell-like structures [58]. Double immunostaining showed that DOR colocalized with the sensory neuron marker calcitonin gene-related peptide (CGRP), a neural peptide that modulates cardiac and blood vessel functions [59]. Exactly which neurotransmitters (and co-transmitters) are released by nerve fibers that express opioid receptors, and if they translate to $QT_C$ prolongation, are important questions for future studies to address.

Another opioid with a large hERG safety margin and clinical QTc prolongation that crosses the 10 msec threshold of regulatory concern is hydrocodone (safety margin for $I_{hERG} > 1238$). This drug is a full agonist at MOR with low affinity for DOR and KOR (https://www. accessdata.fda.gov/drugsatfda_docs/label/2019/206627s010lbl.pdf). The $C_{max}$ of hydrocodone following a 120 mg oral dose is 110 ng/mL or 0.24 μM, converted using a molecular weight of 449.4 g/mol. Based on 33% plasma protein binding, free $C_{max}$ of hydrocodone is 0.081 μM. The hERG channel study report for this drug's application package shows that at 100 μM, hydrocodone inhibited $I_{hERG}$ by 33% (https://www.accessdata.fda.gov/drugsatfda_docs/nda/ 2014/206627Orig1s000PharmR.pdf). The safety margin for hydrocodone is thus >1238. That there are opioids with mechanisms of $QT_C$ prolongation not involving acute hERG channel block has important implications regarding cardiac safety studies for this drug class. During

early discovery, drug candidates are screened for hERG channel block to predict their likelihood of prolonging the $QT_C$ interval. A negative hERG signal for opioids thus may not be predictive of a lack of $QT_C$ prolongation. Pertinent to regulatory review of submitted cardiac safety data, there remain several outstanding questions for $QT_C$ prolongation observed with opioids. What is the extent of $QT_C$ prolongation by opioids in the absence of direct cardiac ion channel interaction, and does it saturate? Opioid receptors are downregulated in the continued presence of agonist. If this form of $QT_C$ prolongation indeed were mediated by opioid receptor activation, then does the $QT_C$ change track with receptor downregulation? How does the $QT_C$ response differ between first-time vs. chronic users of opioids? Regarding opioid use-associated TdP risk, postmarket experience for buprenorphine shows no association with TdP. In contrast, methadone, which directly blocks hERG channels, is associated with high TdP risk. Future studies that assess cardiac ion channel pharmacology, electrocardiographic effects, and postmarket data for additional opioids are important to evaluate whether the lack of association with TdP as seen with buprenorphine can be generalized to other drugs with $QT_C$ prolongation not mediated by acute hERG channel block.

## Acknowledgments

The authors thank Monika Deshpande, PhD, and J. Rick Turner, PhD, DSc, for editorial assistance.

## Author Contributions

**Conceptualization:** Lars Johannesen, David G. Strauss, Wendy W. Wu.

**Data curation:** Phu N. Tran, Jiansong Sheng, Aaron L. Randolph, Claudia Alvarez Baron, Nicolas Thiebaud, Dakshesh Patel, Wendy W. Wu.

**Formal analysis:** Phu N. Tran, Jiansong Sheng, Aaron L. Randolph, Claudia Alvarez Baron, Nicolas Thiebaud, Ming Ren, Min Wu, Dakshesh Patel, Ksenia Blinova, Wendy W. Wu.

**Funding acquisition:** David G. Strauss.

**Investigation:** Phu N. Tran, Jiansong Sheng, Aaron L. Randolph, Claudia Alvarez Baron, Nicolas Thiebaud, Dakshesh Patel, Ksenia Blinova.

**Methodology:** Phu N. Tran, Jiansong Sheng, Aaron L. Randolph, Min Wu, Lars Johannesen, Dakshesh Patel, Wendy W. Wu.

**Resources:** Donna A. Volpe, David G. Strauss.

**Supervision:** Wendy W. Wu.

**Validation:** Wendy W. Wu.

**Visualization:** Ksenia Blinova, Wendy W. Wu.

**Writing – original draft:** Lars Johannesen, Donna A. Volpe, Wendy W. Wu.

**Writing – review & editing:** Phu N. Tran, Aaron L. Randolph, Claudia Alvarez Baron, Nicolas Thiebaud, Ming Ren, Lars Johannesen, Donna A. Volpe, Ksenia Blinova, David G. Strauss, Wendy W. Wu.

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
