## [Decision Letter · Decision Letter 0]

20 Aug 2020

PONE-D-20-22214

Mechanisms of QT prolongation by buprenorphine cannot be explained by direct hERG channel block

PLOS ONE

Dear Dr. Wu,

Thank you for submitting your manuscript to PLOS ONE. After careful consideration, we feel that it has merit but does not fully meet PLOS ONE’s publication criteria as it currently stands. Therefore, we invite you to submit a revised version of the manuscript that addresses the points raised during the review process.

All Reviewers and myself found the manuscript to be of interest, however there needs to be some fine tuning of the language used for clarity, especially in the abstract and in the title of the manuscript. Furthermore, there needs to be some further discussion of compound concentrations and exposure. All additional comments from the Reviewers should also be addressed.

We look forward to receiving your revised manuscript.

Kind regards,

Daniel M. Johnson, PhD

Academic Editor

PLOS ONE

Journal Requirements:

"This work was supported by the operating budget of the Division of Applied Regulatory

Science at the United States Food and Drug Administration."

We note that one or more of the authors are employed by a commercial company: "CiPALab and Vertex Pharmaceuticals (Europe) Ltd "

Reviewers' comments:

Reviewer's Responses to Questions

**Comments to the Author**

1. Is the manuscript technically sound, and do the data support the conclusions?

Reviewer #1: Yes

Reviewer #2: Yes

Reviewer #3: Partly

2. Has the statistical analysis been performed appropriately and rigorously? 

Reviewer #1: Yes

Reviewer #2: Yes

Reviewer #3: Yes

3. Have the authors made all data underlying the findings in their manuscript fully available?

Reviewer #1: Yes

Reviewer #2: Yes

Reviewer #3: Yes

4. Is the manuscript presented in an intelligible fashion and written in standard English?

Reviewer #1: Yes

Reviewer #2: Yes

Reviewer #3: Yes

5. Review Comments to the Author

Reviewer #1: The manuscript by Wu et al., investigates the mechanisms of the QT prolongation seen after administration of buprenorphine in human trials. The effects of buprenorphine and other opioid analogues on key ion channel currents are investigated using the whole cell patch clamp technique and the effects of the opioid agonists/antagonists are also assessed in iPSC-CMs.

The paper is well written and data are clearly presented. I am happy to recommend this for publication but have one main comment and some minor issues that should be addressed beforehand.

It would be more straightforward if all clinical reference concentrations (and affected graphs) were presented as free concentrations rather than the current mixture of free (bup, methadone, naltrexone) and total (norbup, naloxone). I realise the PPB values are not always available. However, the authors describe the simulated PPB value for norbuprenorphine (but do not use it) and a quick search revealed a paper that may be useful for naloxone (Naloxone protein binding in adult and foetal plasma. L. A. Asali & K. F. Brown. European Journal of Clinical Pharmacology volume 27, pages459–463(1984)). Perhaps the missing PPB data could also be generated at a CRO ?

There are a number of MINOR issues that should be addressed prior to publication:

Abstract “…..that repolarizes the ventricular action potentials (APs).” – pedantic point but suggest re-wording this. It is the myocytes that are repolarised.

Intro – “….that is important for ventricular action potential (AP) repolarization”. As above , suggest rewording to something like……”…important during the repolarization phase of the ventricular action potential (AP)”

page 6 – no incubation details ((%C) and temp) for Cav1.2 expressing cells given.

page 13- suggest adding Fredericias’s formula for clarity (or the variation of it that you have used, as Fredericia’s formula is actually for QT)

Page 13 – you have described calculation of FPD parameters but there is no description of the other parameters calculated for the MEA studies – double delta BP and SA.

Results – suggest safety margins do not need to be presented to 1 decimal place (at least for those >10). The same goes for fold margin page 19 eg 1111.1x

Page 19 – worth reiterating what SA refers to as you have done for BP

Page 20 – “For reference, buprenorphine’s total Cmax is 0.022 μM, and simulation

predicted norbuprenorphine…..” Suggest that buprenorphine should actually be norbuprenorphine in this sentence.

I also note that there is inconsistency in the presentation of free Cmax values between drugs eg buprenorphine to 2DP and norbuprenorphine to 3DP. Suggest standardising.

Page 20 “Thus, the in vitro concentrations tested in these studies are likely much higher than the free drug level achieved with therapeutic dosing.” I think it may be worth calculating a margin (accepting that this based on simulated PPB).

Page 20 – Naloxone naltrexone – “No effect on ��SA was observed for either drug at any tested concentration.” It looks like there was a decrease in SA at 100uM for naloxone. Fig 8E

Table 2 – no indication if steady state conc of norbuprenorphine is free or total. This also differs to the value given in Table 1.

Fig 8 legend. Suggest indicating that the vertical dotted line refers to Cmax and indicate that this is sometime free and sometimes total (unless you can standardise this).

Reviewer #2: This manuscript by Wu and colleagues provides a detailed study of the electrophysiologic effects of various opioid agonists and antagonists on multiple cardiac currents (expressed in heterologous systems) as well as human induced pluripotent stem-cell derived cardiomyocytes (pSC-CMs) While carefully performed and results considered in the context of clinical experiences, the findings are not unexpected (not all opioids demonstrate similar electrophysiologic effects). An attempt to explain the multichannel blocking activity on overall repolarization (prolongation or shortening) using an in-silico model would have strengthened the manuscript. Finally, this reviewer was disappointed by the overall conclusion that QTc prolongation with buprenorphine was not explained by any of the findings. It is also uncertain how the possible mechanism suggested (Opioid receptor activation) might apply to all opioid agonists, and not only buprenorphine.

Specific comments:

The abstract leads the reader to believe that buprenorphine causes QTc prolongation. Since QTc prolongation is associated with Torsades, it is expected that buprenorphine would be linked to Torsades. However, the end of the abstract states that the is no evidence that buprenorphine use is associated with TdP. Thus, one is left with either 1) a non-hERG mediated effect eliciting QTc prolongation, or 2) a false-positive QTc finding. Should this second option be considered further?

Reviewer #3: The experimental parts of this manuscript were done in different assays.

Major question: the title of this manuscript is not right based on following reason:

1. The authors have not analyzed the well concentration of buprenorphine in experimental setting, therefore the authors do not know if the dose in the well being so called “clinical free Cmax or Cmax level”. There are different several CNS or pain drugs which caused long QT in man, yet the doses in vitro assay are often much higher than its respectively free Cmax (>10-100 folders than its free Cmax): the reasons are complex (solubility, lipophilic, high in tissue, low in plasma, possible combining with ion trafficking etc..).

2. Doses in MEA assay could be lower than the target doses (in serum medium with protein).

3. LQT and arrhythmias with the drug are often in overdose condition and also long term-use while in vitro assays are very short experiment protocols within minutes or a hour, which is not matching to the conditions in clinic.

4. There are clear effects of buprenorphine on HEREG, Na+ and MEA at so-called high dose in this manuscript. These effects are very similar to other opium agonists such as Dextropropoxyphene or CNS drug Thioridazine , sertindole, ziprasidone etc..

Therefore , strongly suggest that authors to re-write the title and discussion and conclusion.

6. PLOS authors have the option to publish the peer review history of their article (what does this mean?). If published, this will include your full peer review and any attached files.

Reviewer #1: **Yes: **Dr Matt Skinner

Reviewer #2: No

Reviewer #3: No

---

## [Author Response · Author response to Decision Letter 0]

30 Sep 2020

Dear Dr. Johnson,

We thank you and the three reviewers for reading and commenting on our manuscript. Below is our point-to-point response (in blue). We believe that by addressing all the comments, we have significantly strengthened our manuscript. I hope you find it now suitable for publication by PLOS ONE. Please let me know if you have any question.

Sincerely, 

Wendy Wu

Reviewer #1: The manuscript by Wu et al., investigates the mechanisms of the QT prolongation seen after administration of buprenorphine in human trials. The effects of buprenorphine and other opioid analogues on key ion channel currents are investigated using the whole cell patch clamp technique and the effects of the opioid agonists/antagonists are also assessed in iPSC-CMs. The paper is well written and data are clearly presented. I am happy to recommend this for publication but have one main comment and some minor issues that should be addressed beforehand.

It would be more straightforward if all clinical reference concentrations (and affected graphs) were presented as free concentrations rather than the current mixture of free (bup, methadone, naltrexone) and total (norbup, naloxone). I realise the PPB values are not always available. However, the authors describe the simulated PPB value for norbuprenorphine (but do not use it) and a quick search revealed a paper that may be useful for naloxone (Naloxone protein binding in adult and foetal plasma. L. A. Asali & K. F. Brown. European Journal of Clinical Pharmacology volume 27, pages459–463(1984)). Perhaps the missing PPB data could also be generated at a CRO?

We thank Dr. Skinner for sharing the reference describing PPB of naloxone in human plasma. Based on these empirical data, we have calculated the free Cmax for naloxone, and have used this value to calculate the safety margins of this drug for the studied ionic currents. Accordingly, we have updated relevant sections in “Materials and Methods” (p. 15), “Results” (p. 19), “Discussion and Conclusions” (p. 23), Table 1, and Figures 5E and 8 - wherever naloxone’s free Cmax was mentioned or used to calculate the safety margins, or graphically illustrated. 

Regarding norbuprenorphine, we prefer presenting this drug’s total Cmax, and not free Cmax based on simulated results, to reflect the fact that PPB of this metabolite in human plasma has not been empirically determined (as stated on p. 15 of the manuscript). Using norbuprenorphine’s total Cmax, which would produce a smaller hERG safety margin than if we were able to use free Cmax, already eliminated the likelihood of this metabolite contributing to QTC prolongation in the TQT studies of BUTRANS® presented by Harris et al., 2017 via the mechanism of direct and acute hERG channel block. Given that results from experiments to measure PPB of norbuprenorphine would not impact interpretation of our work, we are unable to justify funding these experiments.

There are a number of MINOR issues that should be addressed prior to publication:

Abstract “…..that repolarizes the ventricular action potentials (APs).” – pedantic point but suggest re-wording this. It is the myocytes that are repolarised.

Reworded as Dr. Skinner stated below.

Intro – “….that is important for ventricular action potential (AP) repolarization”. As above , suggest rewording to something like……”…important during the repolarization phase of the ventricular action potential (AP)”

Reworded as Dr. Skinner stated.

page 6 – no incubation details ((%C) and temp) for Cav1.2 expressing cells given.

Incubation details now provided.

page 13- suggest adding Fredericias’s formula for clarity (or the variation of it that you have used, as Fredericia’s formula is actually for QT)

See answer below.

Page 13 – you have described calculation of FPD parameters but there is no description of the other parameters calculated for the MEA studies – double delta BP and SA.

We thank Dr. Skinner for pointing out that clarification of methods used for MEA data analysis is needed. We have now added the Fridericia formula and double delta BP and SA information to the Data analysis for MEA recordings section on p. 13. 

Results – suggest safety margins do not need to be presented to 1 decimal place (at least for those >10). The same goes for fold margin page 19 eg 1111.1x

Done. Now to whole numbers for all drugs except for methadone, which has a safety margin of 6.4.

Page 19 – worth reiterating what SA refers to as you have done for BP

Defined. 

Page 20 – “For reference, buprenorphine’s total Cmax is 0.022 µM, and simulation predicted norbuprenorphine…..” Suggest that buprenorphine should actually be norbuprenorphine in this sentence.

Changed. 

I also note that there is inconsistency in the presentation of free Cmax values between drugs eg buprenorphine to 2DP and norbuprenorphine to 3DP. Suggest standardising.

Thank you for this recommendation of standardizing. We now present free and total Cmax to 2 significant digits. 

Page 20 “Thus, the in vitro concentrations tested in these studies are likely much higher than the free drug level achieved with therapeutic dosing.” I think it may be worth calculating a margin (accepting that this based on simulated PPB).

We prefer using empirical data to calculate free Cmax, hence safety margin for norbuprenorphine was calculated using total Cmax (as is presented in the text). 

Page 20 – Naloxone naltrexone – “No effect on ΔΔSA was observed for either drug at any tested concentration.” It looks like there was a decrease in SA at 100uM for naloxone. Fig 8E

 We thank Dr. Skinner for catching this. We have now clarified that no effects on ΔΔSA was observed for either drug up to 10 µM (p. 20). 

Table 2 – no indication if steady state conc of norbuprenorphine is free or total. This also differs to the value given in Table 1.

This is steady state total Cmax for norbuprenorphine. We have fixed the discrepancy.

Fig 8 legend. Suggest indicating that the vertical dotted line refers to Cmax and indicate that this is sometime free and sometimes total (unless you can standardise this).

We have updated Figure 8 so that naloxone’s vertical dotted lines reflect free Cmax. We have also clarified in Figure 8 legend that the dotted lines for all drugs except norbuprenorphine reflects free Cmax; for norbuprenorphine, total Cmax.

Reviewer #2: This manuscript by Wu and colleagues provides a detailed study of the electrophysiologic effects of various opioid agonists and antagonists on multiple cardiac currents (expressed in heterologous systems) as well as human induced pluripotent stem-cell derived cardiomyocytes (pSC-CMs) While carefully performed and results considered in the context of clinical experiences, the findings are not unexpected (not all opioids demonstrate similar electrophysiologic effects). An attempt to explain the multichannel blocking activity on overall repolarization (prolongation or shortening) using an in-silico model would have strengthened the manuscript. 

We thank Reviewer 2 for this comment. In silico prediction of drug-induced electrical disturbance would need to be considered in the context of clinical exposure levels still. For buprenorphine, norbuprenorphine, naltrexone, and naloxone - drugs with very large safety margins for all ionic currents, exposure near clinical levels would produce no perturbation of ion channel activities hence translate to no electrophysiological changes based on simulation. Therefore, in silico modeling would not alter the conclusions of our study. 

Finally, this reviewer was disappointed by the overall conclusion that QTc prolongation with buprenorphine was not explained by any of the findings. It is also uncertain how the possible mechanism suggested (Opioid receptor activation) might apply to all opioid agonists, and not only buprenorphine.

If mechanisms of QT prolongation were initiated upon activation of opioid receptors, as results from Harris et al., 2017 suggested, then one would expect this mechanism to occur for all opioid receptor agonists. For example, one would hypothesize that QT prolongation by methadone would involve both the indirect mechanism (initiated upon opioid receptor activation) and the direct mechanism of this drug blocking hERG channels directly. 

While we did not identify the cellular mechanisms whereby buprenorphine elicits QT prolongation in the present study, our results are important as they served to close many knowledge gaps in the literature that were mentioned in the “Introduction” and “Discussion” sections. Without filling these knowledge gaps, one would still hypothesize that QT prolongation by buprenorphine to occur by direct drug interaction with cardiac ion channels, which is the most common mechanism published to date. Therefore, our results provide a solid foundation for future research to focus on evaluating indirect and unconventional QT prolongation mechanism by opioids, and the release of original electrophysiology data upon publication allows readers to verify data analysis that led to our conclusions.

Specific comments:

The abstract leads the reader to believe that buprenorphine causes QTc prolongation. Since QTc prolongation is associated with Torsades, it is expected that buprenorphine would be linked to Torsades. However, the end of the abstract states that the is no evidence that buprenorphine use is associated with TdP. Thus, one is left with either 1) a non-hERG mediated effect eliciting QTc prolongation, or 2) a false-positive QTc finding. Should this second option be considered further?

A buprenorphine product indeed has been found to be associated with QTC prolongation in TQT studies (Harris et al., 2017). However, not all QTC prolonging drugs are torsadogenic. Therefore, we performed this study to understand whether buprenorphine shares similar electrophysiological properties as non-torsadogenic, QTC prolonging drugs like ranolazine, amiodarone, and verapamil to understand its risk. Reviewer 2 raised an important point – a potential false positive QTC results for BUTRANS. We find this scenario to be unlikely for the following reasons. First, there were two randomized, positive- and placebo-controlled clinical trials reported in the study. The chance of having both to identify a false positive QTC signal seems low. Second, QTC prolongation associated with buprenorphine was dose- and concentration-dependent. Third, QTC prolongation associated with buprenorphine was not observed when this drug was co-administered with naltrexone, an opioid receptor antagonist. These reasons, coupled with our study results, led us to hypothesize indirect and unconventional mechanism of QTC prolongation for buprenorphine. 

We would like to clarify that we specifically stated that the mechanism is not due to buprenorphine directly blocking hERG channels. We have not ruled out whether or not this drug elicits intracellular signaling cascades within ventricular myocytes that then modulate hERG channel activity to delay ventricular repolarization. Our hypothesis regarding indirect effect of this drug on cardiac ion channels (e.g., by affecting neurotransmission and subsequent activation of neurotransmitter receptor-linked intracellular signaling cascades to delay repolarization) was presented in the manuscript. 

Reviewer #3: The experimental parts of this manuscript were done in different assays. Major question: the title of this manuscript is not right based on following reason:

1. The authors have not analyzed the well concentration of buprenorphine in experimental setting, therefore the authors do not know if the dose in the well being so called “clinical free Cmax or Cmax level”. There are different several CNS or pain drugs which caused long QT in man, yet the doses in vitro assay are often much higher than its respectively free Cmax (>10-100 folders than its free Cmax): the reasons are complex (solubility, lipophilic, high in tissue, low in plasma, possible combining with ion trafficking etc..).

 We thank Reviewer 3 for these comments. We did not perform concentration verification of drug solutions used in the in vitro patch clamp assays and iPSC-CM experiments. Thus, we do not know the actual drug concentrations exposed to cells overexpressing cardiac ion channels and iPSC-CMs. This is an important limitation of this manuscript that we have now pointed out in the “Discussion and Conclusions” section (p. 24, also shown below). 

 “An important limitation of this study is the lack of concentration analysis to confirm actual drug concentrations exposed to the recorded cells. Actual concentrations may differ from nominal drug concentrations due to issues related to stability, solubility, adsorption in the perfusion apparatus, and in the case of MEA experiments, the presence of serum proteins in the medium (51-53). Such possibility has led to the question whether the central conclusion of this study – that buprenorphine and norbuprenorphine do not directly block hERG channels with therapeutic dosing – can still be supported. Two lines of reasoning suggest “yes”. First, despite the uncertainty in in vitro drug concentrations, several retrospective studies have pointed to a hERG safety margin of 30 to 45 and above to identify drugs with low QT prolongation risk due to direct hERG channel block. Buprenorphine’s safety margin is far greater than that threshold and excess of 10,000. Even norbuprenorphine’s safety margin, determined using total rather than free Cmax, is excess of 1,600. Second, a previous study that compared drug loss for 39 compounds showed the most extensive loss to be 74% (51). Even with that level of drug loss, buprenorphine’s safety margin would still be 2600, whereas that for norbuprenorphine determined using total Cmax would still be 416. Therefore, the lack of concentration analysis in this study may affect the actual safety margin values for buprenorphine and norbuprenorphine. However, due to the magnitude of these values, the study conclusion - that acute and direct block of hERG channels by buprenorphine (and norbuprenorphine) is unlikely to occur with therapeutic dosing and cannot explain QT prolongation seen in Harris et al., 2017 (1) – is not altered.”

 Reviewer alluded to the possibility of tissue accumulation for drug. In this case, free Cmax determined using plasma would be inappropriate to approximate exposure level that could lead to hERG channel interaction, and the use of free Cmax would result in a much larger hERG safety margin then if tissue level were used. However, buprenorphine has been studied in rat and sheep ventricular myocytes where accumulation could presumably occur. In sheep Purkinje fibers and rat papillary muscles, concentrations of buprenorphine required to reduce AP height, reduce maximum rate of rise (Vmax), and prolong APD were still in the micromolar concentration range (Boachie-Ansah et al., 1989), far higher than the subnanomolar clinical exposure level. Tissue accumulation is thus unlikely to explain the large safety margin for buprenorphine.

 Reviewer also raised the possibility of hERG channel trafficking mediating QT prolongation. This mechanism is not due to direct drug block of hERG channels, as stated in our title for this manuscript. We agree that our patch clamp studies were not designed to evaluate trafficking mechanism. Clinical data in Harris et al., 2017 showed that co-administration of naltrexone prevented QT prolongation by buprenorphine. Whatever the underlying mechanism of QT prolongation not due to acute and direct hERG channel block, it seems likely that it is initiated by opioid receptor activation.

2. Doses in MEA assay could be lower than the target doses (in serum medium with protein).

Presence of serum in the recording medium can alter the effective drug concentration, with the direction of change in a drug-specific manner. Binding of the drug to the serum proteins may decrease the effective drug concentration, while increased drug solubility in the presence of the serum proteins may lead to the increased effective drug concentration as compared to a serum-free media. These effects have been examined in our prior study (Schocken et al., 2018). We have added this point to the limitation section in the “Discussion and Conclusions” (p. 24). 

3. LQT and arrhythmias with the drug are often in overdose condition and also long term- use while in vitro assays are very short experiment protocols within minutes or a hour, which is not matching to the conditions in clinic.

 In overdose scenario, systemic level of opioids could increase to a level that leads to direct drug interaction with hERG channels, as we have reported for loperamide (Sheng et al., 2017). However, this possibility does not explain buprenorphine induced QT prolongation reported by Harris et al., 2017.

 If chronic exposure is necessary to induce QT prolongation, and QT change does not match PK profile of the drug, then this also argues against direct hERG channel block as the underlying mechanism.

4. There are clear effects of buprenorphine on HEREG, Na+ and MEA at so-called high dose

in this manuscript. These effects are very similar to other opium agonists such as Dextropropoxyphene or CNS drug Thioridazine , sertindole, ziprasidone etc..

Therefore , strongly suggest that authors to re-write the title and discussion and conclusion.

We thank Reviewer 3 for this comment. We and others have found that buprenorphine’s effects on cardiac ion channels and cardiac action potentials require drug concentration in the micromolar range – a level that is far above what is achieved with clinical dosing (steady state free Cmax of buprenorphine is < 1 nM; see “Materials and Methods”). For hERG current, the ratio between IC50 and free Cmax was in excess of 10000. Therefore, we concluded that this drug’s QT prolongation is not mediated by the drug molecule acutely and directly blocking hERG channels – a mechanism that is shared by all torsadogenic and QT prolonging drugs withdrawn from the market. We have further tested buprenorphine’s major metabolite, norbuprenorphine, on a panel of cardiac ion channels and iPSC-CMs, and ruled out the possibility that this metabolite blocks hERG channels acutely and directly with clinical dosing (the ratio of IC50 and total Cmax was greater than 1600). Since buprenorphine’s QT prolongation was not observed when this drug was co-administered with an opioid receptor antagonist naltrexone, the most parsimonious explanation is that this drug’s QT prolongation is mediated through opioid receptor activation. 

Reviewer 3 stated that 4 drugs - dextropropoxyphene (or propoxyphene), thioridazine, sertindole, and ziprasidone - are like buprenorphine. We are unclear what electrophysiology features the Reviewer was using for this comparison. Our analysis, shown below, indicates that QT prolongation and incidence of arrhythmia by all 4 drugs likely involve acute and direct block of hERG channels, and that these drugs’ electrophysiological features are therefore distinct from those of buprenorphine (and norbuprenorphine). 

1) Propoxyphene is an MOR associated with PR, QRS, QTC prolongation and TdP (https://www.fda.gov/drugs/drug-safety-and-availability/fda-drug-safety-communication-fda-recommends-against-continued-use-propoxyphene#references) (Adler et al., 2011). Propoxyphene’s IC50 for hERG current is 44.7 µM, and its major metabolite norpropoxyphene has an IC50 for hERG current of 33.2 µM (Ulens et al., 1999). Importantly, in the micromolar concentration range, both the parent and metabolite not only block hERG channels, but additionally increase the channels’ Na+ permeability. This latter effect would likely reduce the repolarizing influence due to hERG channel activation, since the reversal potential for hERG current would be depolarized. Propoxyphene’s Cmax following a single 65 mg oral dose is 0.1 �g/mL or 0.294 µM, based on a molecular weight of 339.47 g/mol (https://www.drugs.com/pro/propoxyphene-capsules.html). Given 80% plasma protein binding, free Cmax of propoxyphene is 0.059 µM, and the safety margin of this drug for hERG current is 759. Norpropoxyphene’s Cmax for is 0.2 �g/mL or 0.615 µM following a single dose of 65 mg, converted using a molecular weight of 325.4 g/mol. Protein binding of 75% have been reported for norpropoxyphene (Giacomini et al. 1980). The safety margin of norpropoxyphene is only 216. Note that safety margins of both propoxyphene and norpropoxyphene are much lower than those of buprenorphine and norbuprenorphine, and are conservatively estimated. The estimates are conservative because they are based on Cmax values following a single dose only. Accumulation of both the parent and metabolite is expected with a dosing regimen of every 4 hours and reported half-lives of 6-12 hrs and 30-36 hrs for propoxyphene and norpropoxyphene respectively, consistent with reported maximum concentrations of up to 0.85 ug/mL (propoxyphene) and 3 ug/mL (norpropoxyphene) with chronic use (https://www.citizen.org/article/petition-to-ban-all-propoxyphene-darvon-products/#_edn31). If steady state Cmax values were used to calculate safety margins of propoxyphene and norpropoxyphene, then safety margins for these two molecules would become smaller. For example, safety margins derived using the abovementioned maximum concentrations (accounting for protein binding) would be 262 for propoxyphene and 44 for norpropoxyphene. Direct hERG channel block following propoxyphene use due to its metabolite thus cannot be ruled out. 

2) Ziprasidone is an antipsychotic agent. For its Cmax, we used data from Geodon®. At a maximum dose of 80 mg BID, steady state Cmax of ziprasidone is 202 ng/mL (page 28 of ClinPharm review). Based on a molecular weight of 412.94 g/mol (for free base), Cmax converts to 0.49 µM. Given 99% protein binding, ziprasidone’s free Cmax is 5 nM. Zaprasidone’s IC50 for hERG current is 120 nM (Su et al., 2006). The safety margin of this drug for hERG current is 24, much smaller than the safety margins of buprenorphine and norbuprenorphine. Based on the close proximity between in vitro IC50 and clinical exposure levels, that ziprasidone may reach a systemic level with clinical dosing to block hERG channels directly cannot be ruled out.

3) We were unable to identify drug label information for thioridazine, a first generation antipsychotic drug that was withdrawn because it caused severe cardiac arrhythmias. Therefore, information from the literature was used. The most potent IC50 for thioridazine on hERG current is 33 nM (Crumb et al., 2013). Redfern et al., 2003 (Supplemental data, p. 20) reported that the free Cmax of this drug ranged from 208 to 979 nM. Thus, the safety margin for of this drug is 0.03 to 0.15, much lower than the safety margins for buprenorphine and norbuprenorphine and suggests the possibility of direct drug interaction with hERG channels with clinical dosing.

4) Sertindole is an antipsychotic medication. The IC50 of this drug is protocol-dependent, ranging from 3 to 14 nM (Rampe at al., 1998). Free Cmax of this drug ranged from 0.2 to 1.59 nM. Using the more potent IC50 value, the safety margin of this drug is estimated to be 1.9 to 15, much lower than those of buprenorphine and norbuprenorphine and suggests the possibility of direct drug interaction with hERG channels with clinical dosing. 

The 4 drugs that Reviewer 3 mentioned are therefore likely to affect cardiac repolarization by directly blocking hERG channels. These 4 drugs are similar to methadone that was also studied in the present manuscript and differ from buprenorphine and norbuprenorhine. 

We believe that our title is supported by our data and rationale provided in this rebuttal letter. Hence, it is not changed. We have updated our “Discussion and Conclusions” section to highlight limitation of the present study per Reviewer’s comment.

6. PLOS authors have the option to publish the peer review history of their article (what does this mean?). If published, this will include your full peer review and any attached files.

Do you want your identity to be public for this peer review? For information about this choice, including consent withdrawal, please see our Privacy Policy.

Reviewer #1: Yes: Dr Matt Skinner Reviewer #2: No

Reviewer #3: No

---

## [Decision Letter · Decision Letter 1]

14 Oct 2020

Mechanisms of QT prolongation by buprenorphine cannot be explained by direct hERG channel block

PONE-D-20-22214R1

Dear Dr. Wu,

We’re pleased to inform you that your manuscript has been judged scientifically suitable for publication and will be formally accepted for publication once it meets all outstanding technical requirements.

Kind regards,

Daniel M. Johnson, PhD

Academic Editor

PLOS ONE

Additional Editor Comments (optional):

Reviewers' comments:

Reviewer's Responses to Questions

**Comments to the Author**

1. If the authors have adequately addressed your comments raised in a previous round of review and you feel that this manuscript is now acceptable for publication, you may indicate that here to bypass the “Comments to the Author” section, enter your conflict of interest statement in the “Confidential to Editor” section, and submit your "Accept" recommendation.

Reviewer #1: All comments have been addressed

Reviewer #2: All comments have been addressed

Reviewer #3: All comments have been addressed

2. Is the manuscript technically sound, and do the data support the conclusions?

Reviewer #1: (No Response)

Reviewer #2: Yes

Reviewer #3: Yes

3. Has the statistical analysis been performed appropriately and rigorously? 

Reviewer #1: (No Response)

Reviewer #2: Yes

Reviewer #3: Yes

4. Have the authors made all data underlying the findings in their manuscript fully available?

Reviewer #1: (No Response)

Reviewer #2: Yes

Reviewer #3: Yes

5. Is the manuscript presented in an intelligible fashion and written in standard English?

Reviewer #1: (No Response)

Reviewer #2: Yes

Reviewer #3: Yes

6. Review Comments to the Author

Reviewer #1: (No Response)

Reviewer #2: This manuscript has been revised appropriately.

The information provided should be appreciated by readers of PLOSOne.

Reviewer #3: the aurthors now made responses my my comments in details with detail scientific infornations.

The manuscript is now well written.

7. PLOS authors have the option to publish the peer review history of their article (what does this mean?). If published, this will include your full peer review and any attached files.

Reviewer #1: **Yes: **Dr Matt Skinner

Reviewer #2: No

Reviewer #3: No

---

## [Editor Report · Acceptance letter]

28 Oct 2020

PONE-D-20-22214R1 

Mechanisms of QT prolongation by buprenorphine cannot be explained by direct hERG channel block 

Dear Dr. Wu:

I'm pleased to inform you that your manuscript has been deemed suitable for publication in PLOS ONE. Congratulations! Your manuscript is now with our production department. 

Kind regards, 

on behalf of

Dr. Daniel M. Johnson 

Academic Editor

PLOS ONE